# PCP and Wnt pathway components act in parallel during zebrafish mechanosensory hair cell orientation

Joaquin Navajas Acedo [1], Matthew G. Voas[1], Richard Alexander[1], Thomas Woolley [2], Jay R. Unruh [1], Hua Li[1], Cecilia Moens[3] & Tatjana Piotrowski [1]

Planar cell polarity (PCP) plays crucial roles in developmental processes such as gastrulation, neural tube closure and hearing. Wnt pathway mutants are often classified as PCP mutants due to similarities between their phenotypes. Here, we show that in the zebrafish lateral line, disruptions of the PCP and Wnt pathways have differential effects on hair cell orientations. While mutations in the PCP genes *vangl2* and *scrib* cause random orientations of hair cells, mutations in *wnt11f1*, *gpc4* and *fzd7a/b* induce hair cells to adopt a concentric pattern. This concentric pattern is not caused by defects in PCP but is due to misaligned support cells. The molecular basis of the support cell defect is unknown but we demonstrate that the PCP and Wnt pathways work in parallel to establish proper hair cell orientation. Consequently, hair cell orientation defects are not solely explained by defects in PCP signaling, and some hair cell phenotypes warrant re-evaluation.

[1] Stowers Institute for Medical Research, Kansas City, MO, USA. [2] Cardiff School of Mathematics, Cardiff University, Senghennydd Road, Cardiff CF24 4AG, UK. [3] Fred Hutchinson Cancer Research Center, Basic Sciences Division, Seattle, WA, USA. Correspondence and requests for materials should be addressed to T.P. (email: pio@stowers.org)

Cell polarity is crucial for the function of many tissues and its establishment during development has fascinated many generations of biologists. In addition to the well-studied apico-basal polarity of cells, cells are also coordinately aligned in the plane of a tissue's axis, termed planar cell polarity (PCP). PCP relies on the asymmetric distribution of core PCP components, such as Van Gogh/Vangl, Frizzled, Disheveled, Prickle/Spiny-Legs, and Fmi/Celsr and Diego[1–4]. This pathway was discovered in insects[5–8] and subsequently also identified in vertebrates[9–11]. PCP is required during key developmental processes that shape the embryo, such as gastrulation and neural tube closure[1–3], however, how PCP is initiated is poorly understood. One proposed mechanism is that Wnt gradients act as instructive morphogens that initiate the molecular asymmetry of PCP components[12–17]. However, in some contexts Wnt ligands are required but not instructive (permissive) to drive PCP-dependent processes[18–20]. In this study, we set out to investigate the role of Wnt signaling in polarizing sensory hair cells.

The vertebrate inner ear is a classical model to study the function of the PCP pathway, as stereocilia bundle coordination is very sensitive to changes in PCP[21,22]. Wnt gradients have been implicated in establishing PCP in the ear[23,24] but due to the inaccessibility of the ear, the study of the function of the PCP and Wnt signaling pathways in coordinating hair cell alignment is challenging to investigate.

A more experimentally accessible model to study hair cell orientation is the sensory lateral line system of aquatic vertebrates that detects water movements across the body of the animal[25,26]. Because of its superficial location in the skin, the lateral line is amenable to experimental manipulations and live imaging[27]. The lateral line system consists of volcano-shaped sensory organs (neuromasts) that are composed of mantle cells on the outside and support cells and mechanosensory hair cells in the center (Fig. 1a, b [25]). The mechanosensory hair cells are homologous to the ones found in the inner ear[28,29]. The sensory organs are derived from several neurogenic, cephalic placodes/primordia that either migrate into the trunk or into the head[30]. As they migrate, primordia periodically deposit clusters of cells that differentiate into sensory organs[31–33]. PrimordiumI (primI) and primordiumII (primII) both migrate into the trunk but arise from different placodes (primary placode and D0 placode, respectively). The D0 placode also gives rise to a third primordium that migrates onto the dorsal side of the trunk, called primD[25,34].

Hair cells possess short actin-rich stereocilia adjacent to a tubulin-rich, long kinocilium. Each hair cell is planar polarized with the long microtubule-based kinocilium localized to one pole of the cuticular plate (Fig. 1b). Within a sensory organ, hair cells arise in pairs with their kinocilia pointing toward each other but are aligned along a common axis (Fig. 1c). The positioning of the kinocilium along this axis is controlled by Notch signaling/Emx2 and its loss causes all hair cells within a neuromast to point into the same direction[35,36]. PrimI and primII-derived hair cells show different axial polarities, allowing the animal to sense water flow from different directions (Fig. 1c [37]). In primI-derived trunk neuromasts, hair cells are planar polarized along the anterioposterior (A-P) axis and hair cells in primII-derived neuromasts are aligned along the dorsoventral (D-V) axis. The differential hair cell orientation has been correlated with the different directions of primI and primII migration, however, no underlying mechanism has been identified[25,34,37]. At the single hair cell level, previous reports have suggested that PCP regulates hair cell progenitor orientation by controlling cell division angles and cell rearrangements[38,39]. Yet the molecular code for how different neuromast axial orientations and individual hair cell orientations are coordinated remains ill understood.

Here, we show that loss of PCP and Wnt pathway genes (wnt11 (wnt11f1) (formerly known as wnt11r, wnt11-related[40]), gpc4 and fzd7a/7b) have different consequences on hair cell orientations. While mutations in the PCP genes vangl2 and scrib cause disorganized hair cell orientations in all neuromasts, mutations in the Wnt pathway genes wnt11 (wnt11f1), gpc4 and fzd7a/7b show a striking concentric pattern of hair cell orientation in only primII neuromasts. As neither the core PCP component Vangl2, nor Notch/Emx2 signaling are affected in Wnt pathway mutants we conclude that the Wnt pathway acts in parallel to these pathways. In addition, the concentric hair cell phenotype in Wnt pathway mutants is caused by the disruption of coordinated organization of the surrounding support cells, rather than by affecting the axis of polarity or kinocilium positioning in individual hair cells.

The expression patterns of Wnt pathway genes suggest that the Wnt pathway acts very early in lateral line development. Thus, Wnt signaling does not instruct PCP, but acts to coordinate support cell organization during the formation and migration of the primordium before the appearance of hair cells. The molecular mechanisms by which Wnt signaling coordinates support cell orientation remains to be elucidated. Overall, our findings demonstrate that hair cell orientation defects cannot solely be attributed to defects in the PCP pathway and that some phenotypes formerly characterized as PCP defects need to be re-evaluated.

## Results

**Wnt and PCP genes cause different hair cell orientation phenotypes.** During a large in situ screen, we unexpectedly observed asymmetric expression of wnt11 (wnt11f1), the ortholog of mammalian WNT11[40]. wnt11 (wnt11f1) is expressed along the anterior edge of only primI-derived neuromasts, but is absent from primII-derived neuromasts (Fig. 1d, Supplementary Fig. 1). Since Wnt ligands can instruct planar polarization of cells[10,16,17,41–44], we hypothesized that wnt11 (wnt11f1) establishes hair cell orientation by directing PCP in primI-derived neuromasts. We measured hair cell orientation in the cuticular plate using Phalloidin, which labels actin-rich stereocilia but not the tubulin-rich kinocilium (Fig. 1b). We used the kinocilium position to determine the axis of polarity of each hair cell. Phalloidin stainings of sibling primI-derived neuromasts show that hair cells possess a significant orientation bias parallel to the A-P axis based on the angles with respect to the horizontal in rose diagrams (Fig. 1e). In contrast, primII-derived neuromasts show an orientation bias along the D-V axis (Fig. 1k). Furthermore, neighboring hair cells in both primordia show coordinated polarities (Supplementary Fig. 1c, i). Unexpectedly, zygotic and maternal zygotic (MZ) mutations in wnt11 (wnt11f1) do not affect hair cell orientation in primI-derived neuromasts in which wnt11 (wnt11f1) is expressed (Fig. 1f; Supplementary Fig. 1d), but disrupt the hair cell orientation in primII-derived neuromasts (Fig. 1l; Supplementary Fig. 1j). Even though zygotic wnt11 (wnt11f1) mutants also show the phenotype, we from here on used MZwnt11 (wnt11f1) larvae to increase the number of mutant fish for our studies.

gpc4 (glypican4, a heparan sulfate proteoglycan (HSPG)) and fzd7 (a frizzled-class receptor) interact with Wnt ligands during convergent extension (CE), and we tested whether mutations in these genes also cause hair cell polarity defects[18,45–51]. Indeed, gpc4 and MZfzd7a/7b mutants show the same concentric hair cell phenotype as MZwnt11 (wnt11f1) mutants suggesting they act in the same pathway (Fig. 1g, m; and h, n; Supplementary Fig. 1e, f, k, l). This interpretation is supported by the finding that wnt11 (wnt11f1);gpc4 double mutants possess the same phenotype as the

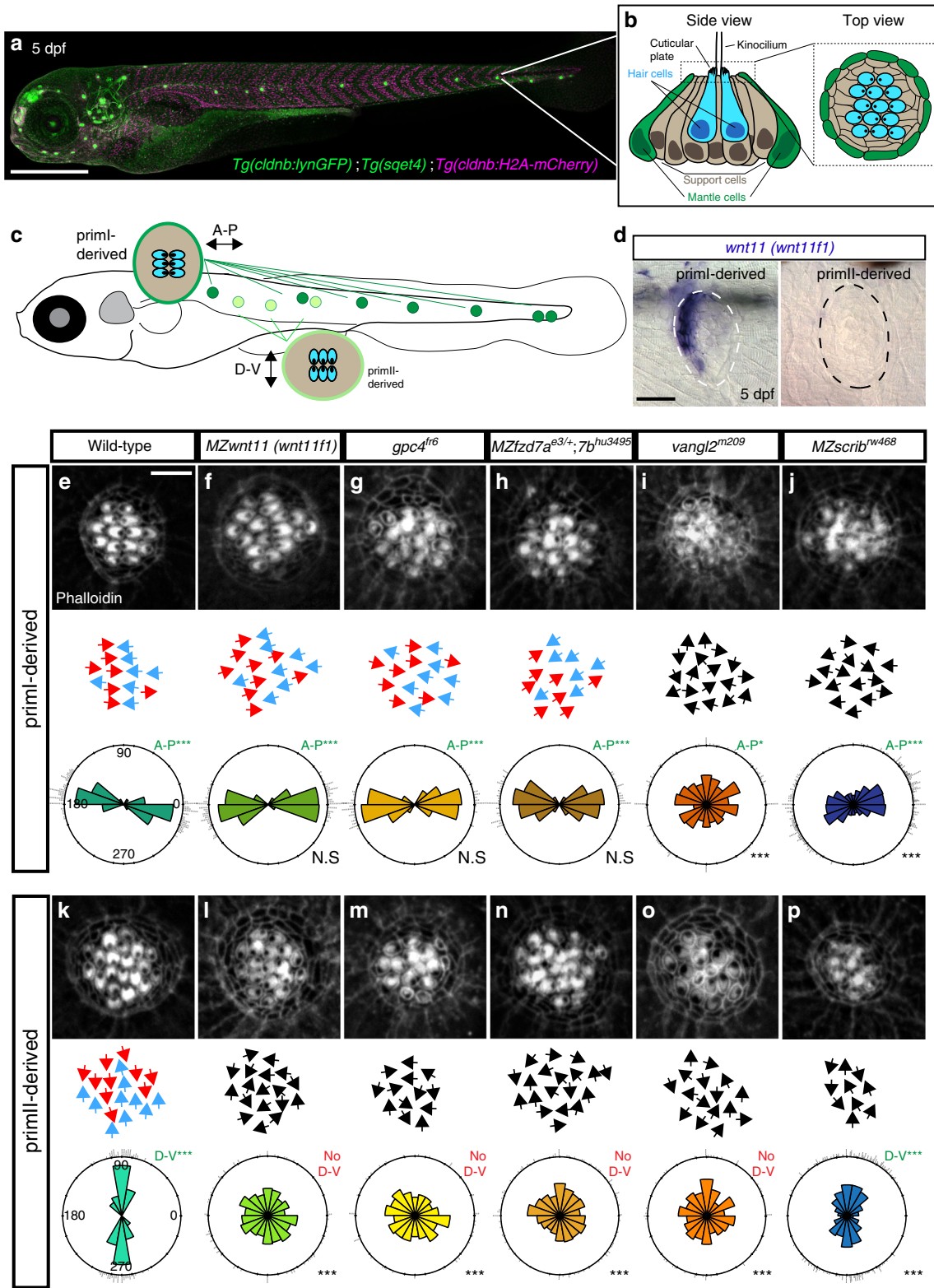

single mutants (Supplementary Fig. 1o). *gpc4* and *fzd7a/7b* also interact with *wnt11f2* (formerly called *wnt11/silberblick*,[40]) during CE[18,50,51]. We, therefore, wondered if *wnt11f2* mutants also possess hair cell defects and if *wnt11f2* possibly interacts with *wnt11 (wnt11f1)*. However, *wnt11f2* mutants have normal hair cell orientations, as do *wnt11 (wnt11f1);wnt11f2* double heterozygous animals, indicating that these two paralogs do not interact (Supplementary Fig. 1p).

Since *gpc4* and *MZfzd7a/7b* mutants have been described as PCP signaling mutants in other contexts[45,46,52,53], we compared the phenotypes of *MZwnt11 (wnt11f1)*, *gpc4* and *MZfz7a/7b* mutants to the hair cell phenotype of the PCP mutants *vangl2* and *scribble (scrib)*. A mutation in the core PCP gene *vangl2* disrupts hair cell orientation in both primI and primII-derived neuromasts[38,39] (Fig. 1i, o; Supplementary Fig. 1g, m). Scribble1 interacts with the PCP pathway in the mouse cochlea and its loss

**Fig. 1** Mutations in the Wnt and PCP pathways affect the hair cell alignment in the lateral line. **a** Confocal image of a 5 days post fertilization (dpf) *Tg(cldnb: lynGFP); Tg(sqet4); Tg(cldnb:H2A-mCherry)* larva. **b** Schematic lateral view of a 5 dpf neuromast showing the different cell types. **c** Diagram of a 5 dpf larva showing the two different orientations of primI and primII-derived hair cells. **d** In situ hybridization of *wnt11 (wnt11f1)* in primI and primII-derived 5 dpf neuromasts. **e–j** Phalloidin stainings show hair cell orientations in primI-derived neuromasts of wild type (**e**), Wnt pathway mutants (**f–h**) and PCP mutants (**i–j**; Fisher's Exact Test $p$-val for *vangl2* primI = $7.33 \times 10^{-28}$, *MZscrib* primI = $1.41 \times 10^{-17}$). Individual hair cell orientation is depicted for each of the conditions tested. Black arrows denote disruption of the wild-type orientation. The Rose diagrams show the hair cell orientation distribution with respect to the longitudinal axis of the animal (horizontal) (WT $n = 194$ hair cells, *MZwnt11 (wnt11f1)* $n = 353$, *gpc4* $n = 222$, *MZfzd7a/7b* $n = 74$, *vangl2* $n = 226$, *MZscrib* $n = 535$). Bottom right depicts the Fisher's exact test comparison with respect to the wild type for each condition. Top right shows the binomial test for each condition. **k–p** Phalloidin staining shows hair cell orientation in primII-derived neuromasts of wild type (**k**), Wnt pathway mutants (**l–n** Fisher's Exact Test $p$-val for *MZwnt11 (wnt11f1)* primII = $1.69 \times 10^{-23}$, *gpc4* primII = $6.93 \times 10^{-31}$, *MZfzd7a/7b* primII = $3.91 \times 10^{-33}$) and PCP mutants (**o–p**, *vangl2* primII $p = 2.52 \times 10^{-21}$, *MZscrib* primII = $9.22 \times 10^{-12}$). Individual hair cell orientation is depicted for for each of the conditions tested and the color code is the same as in (**e–j**). The Rose diagrams reflect the same as in (**e–j**) (WT $n = 209$ hair cells, *MZwnt11 (wnt11f1)* $n = 295$, *gpc4* $n = 214$, *MZfzd7a/7b* $n = 182$, *vangl2* $n = 141$, *MZscrib* $n = 392$). For **e–p**, comparisons of angle distributions with respect to wild type, Fisher's Exact Test ***$p < 0.001$, N.S not significant, binomial test A-P ***$p < 0.001$, A-P* = $p < 0.05$. e"–p". A-P anteroposterior orientation, D-V dorsoventral orientation. Scale bar in **a** equals 500 μm, **d** equals 20 μm, **e** equals 5 μm

causes hair cell defects[54,55]. Likewise, we observed that *MZscrib* zebrafish mutants show a significant deviation of hair cell orientations in both primI and primII-derived neuromasts (Fig. 1j, p). However, the phenotype is not as severe as in *vangl2* mutants and primI and primII neuromasts still show a bimodal distribution along the A-P and D-V axes, respectively (Fig. 1j, p; Supplementary Fig. 1h, n). Therefore, *scrib* likely acts partially redundant with other PCP proteins. These results suggest that while PCP genes are required to establish proper hair cell orientation in all neuromasts, *wnt11 (wnt11f1)*, *gpc4* and *fzd7a/7b* (from now on referred to as Wnt pathway genes), affect hair cell orientation only in primII-derived neuromasts.

Mutations in both Wnt and PCP pathway genes result in misorientation of hair cells. However, we found that mutations in Wnt pathway genes cause hair cells in primII-derived neuromasts to orient in a striking concentric fashion (Fig. 2b, g; Supplementary Fig. 2b–f), whereas in *vangl2* mutants hair cells are oriented randomly (Fig. 2c, h; Supplementary Fig. 1g, m). Likewise, hair cells in the other PCP signaling mutant *MZscrib* do not arrange in a concentric fashion (Supplementary Fig. 2d, g). This difference in phenotype suggests that the Wnt and PCP signaling pathways control hair cell orientation in parallel, rather than through a common pathway. However, Wnt and PCP pathways have been described to interact genetically in the establishment of hair cell orientation in the inner ear[24]. To assess a possible genetic interaction between *vangl2* and *wnt11 (wnt11f1)* in lateral line neuromasts we generated double homozygous *vangl2* and *MZwnt11 (wnt11f1)* larvae. In double homozygous fish, primI- and II-derived neuromasts show random hair cell orientation (Fig. 2e, j; Supplementary Fig. 2h, i). In addition, the concentric phenotype shown by primII neuromasts of *MZwnt11 (wnt11f1)* single mutants (Fig. 2i; Supplementary Fig. 2j) disappears in the double *MZwnt11 (wnt11f1);vangl2* mutants (Fig. 2j; Supplementary Fig. 2k). Due to the genetic compensation between Vangl2 and Vangl1 in the mammalian inner ear[56,57], we tested if *vangl1* and *vangl2* act redundantly during zebrafish lateral line hair cell orientation. Indeed, *vangl1* is expressed in the migrating primI and a few mantle cells in 32 and 50 h post fertilization (hpf) neuromasts (Supplementary Fig. 2l). A t-SNE plot of 5 dpf embryos also shows that a few mantle cells are positive (Supplementary Fig. 2m). However, *vangl1* is not expressed in the migrating primII (Supplementary Fig. 2l). We generated *vangl1* CRISPants which are characterized by shortened, curly tails and notochord defects reminiscent of PCP mutants (Supplementary Fig. 2o [11,45,58,59]). However, these embryos do not show a disruption in their hair cell orientation (Supplementary Fig. 2p). In addition, neither *MZwnt11 (wnt11f1)* or *vangl2* mutants injected with *vangl1* CRISPR show a deviation from the

phenotypes observed in uninjected *MZwnt11 (wnt11f1)* or *vangl2* mutants (Supplementary Fig. 2q, r; Fig. 2b, c, g, h). These findings suggest that *vangl1* and *vangl2* do not act redundantly during the establishment of hair cell orientation in the lateral line. In sum, the disorganized hair cell phenotype caused by a mutation in *vangl2* is epistatic over the concentric phenotype observed in *wnt11 (wnt11f1)* embryos, providing additional support for the hypothesis that the PCP and Wnt pathways work in parallel and have different roles during the establishment of hair cell orientation in primII-derived neuromasts.

**Vangl2 localization is not affected in Wnt pathway mutant hair cells.** Since loss of *vangl2* disrupts the concentric phenotype, we hypothesized that PCP signaling might be correctly established in hair cells of the Wnt pathway mutants. To test whether one of the landmarks of PCP signaling, asymmetric distribution of Vangl2[3], is disrupted in hair cells of Wnt pathway mutants we performed anti-Vangl2 antibody stainings in 5 dpf neuromasts (Fig. 3). In wild type neuromasts, Vangl2 is asymmetrically enriched in approximately 90% of hair cells. Vangl2 is localized to the posterior side of hair cells in primI-derived neuromasts (Fig. 3a) and to the ventral side of hair cells in primII-derived neuromasts (Fig. 3e, i). However, as hair cells in a wild type neuromast possess two different orientations (Fig. 1b, c) only approximately half of the hair cells show Vangl2 signal in the same pole as the kinocilium (Supplementary Fig. 3a), which agrees with previous reports using a fluorescent reporter or a Vangl2 antibody[36,39].

While Vangl2 staining is not detected in *vangl2* mutants (Supplementary Fig. 3b–e), *MZscrib* mutant fish show reduced signal and only 26.3% of hair cells with asymmetric Vangl2 in primI and primII-derived neuromasts (Fig. 3b, f, i) compared with their siblings (Fig. 3c, g, i). The reduction in signal, rather than the complete loss, agrees with reports in the mouse cochlea[55] and might explain why hair cell disorganization in *MZscrib* mutants is not as dramatic as in *vangl2* mutants. In addition, among the 26.3% of hair cells that that show asymmetric Vangl2 in *MZscrib* mutants, approximately 10% of hair cells show randomized Vangl2 localization that is not correlated with the position of the kinocilium (Supplementary Fig. 3a). Together with the finding that only 26.3% of *MZscrib* hair cells show asymmetric distribution of Vangl2, this indicates that PCP signaling is affected in *MZscrib*. In contrast to the PCP mutants, *MZwnt11 (wnt11f1)* mutants possess asymmetric distribution of Vangl2 in hair cells of both primI and primII-derived neuromasts showing that PCP signaling is not affected (Fig. 3d, h, i; Supplementary Fig. 3a). To further test whether hair cells in *MZwnt11 (wnt11f1)* mutants show asymmetric Vangl2

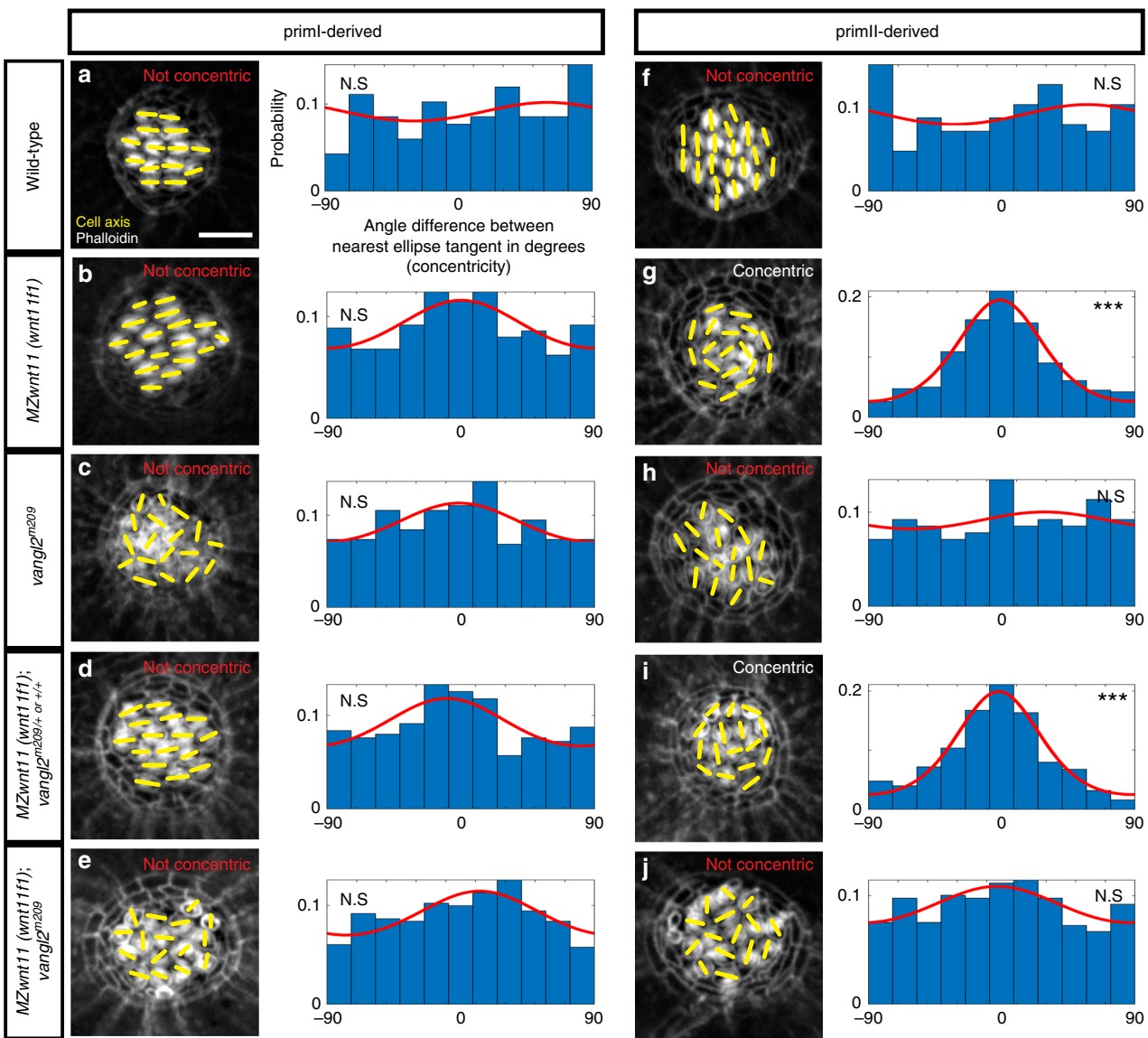

**Fig. 2** PCP and Wnt pathway mutants possess different hair cell phenotypes. In PCP mutants hair cells are randomly oriented, whereas hair cells in primII of Wnt pathway mutants show a concentric orientation pattern. **a–e** Phalloidin images show the cell polarity axis (yellow lines) in primI-derived neuromasts of wild type (**a**), Wnt pathway mutant *MZwnt11 (wnt11f1)* (**b**), PCP mutant *vangl2* (**c**), single *MZwnt11 (wnt11f1)* siblings (**d**), and double *MZwnt11 (wnt11f1); vangl2* larvae (**e**) homozygous mutant hair cells and their distribution of angles with respect to the nearest ellipse tangent (concentricity). Note that none of the conditions shows significant concentricity (uniform distribution *p*-values in a = 0.423, b = 0.054, c = 0.464, d = 0.165, e = 0.077, f = 0.019; WT *n* = 194 hair cells, *MZwnt11 (wnt11f1)* *n* = 353, *vangl2* *n* = 226, *MZwnt11 (wnt11f1);vangl2* sibling *n* = 276, *MZwnt11 (wnt11f1);vangl2* doubles *n* = 406). The Von Mises distributions are shown in red for visual display of the data distribution. **f–j** Phalloidin images show the cell polarity axis (yellow lines) in primII-derived neuromasts of wild type (**f**), Wnt pathway mutant *wnt11 (wnt11f1)* (**g**), PCP mutant *vangl2* (**h**), single *MZwnt11 (wnt11f1)* siblings (**i**) and double *MZwnt11 (wnt11f1);vangl2* (**j**) homozygous mutants and their distribution of angles with respect to the nearest ellipse tangent (concentricity). Note that only the Wnt pathway mutants (**g, i**) show significant concentricity (Uniform distribution *p*-values in f = 0.314, g = $1.28 \times 10^{-27}$, h = 0.86, i = $02.92 \times 10^{-19}$, j = 0.298; WT *n* = 209 hair cells, *MZwnt11 (wnt11f1)* *n* = 295, *vangl2* *n* = 141, *MZwnt11 (wnt11f1);vangl2* sibling *n* = 252, *MZwnt11 (wnt11f1);vangl2* doubles *n* = 365). The von Mises distributions are shown in red for visual display of the data distribution. Yellow lines in **a–j** indicate the hair cell polarity axis, determined by the position of the kinocilium. Not Concentric vs Concentric labels were based on calculations in (**a–j**). Scale bar equals 5 μm

localization, we generated a hair cell-specific promoter-driven GFP-Vangl2 *Tg(myo6b:GFP-XVangl2)*. We then calculated GFP fluorescence enrichment around the circumference (−180° to 180°) of the cuticular plate at the base of the stereo- and kinocilia with 0° on the *x*-axis being the point of maximum intensity (Fig. 3j–o). In wild-type larvae mosaic expression of fluorescent Vangl2 is asymmetrically enriched in hair cells of primI and primII-derived neuromasts (Fig. 3j–l, p). In *MZwnt11 (wnt11f1)* mutants, GFP-Vangl2 is asymmetrically localized in primI and primII-derived neuromasts like in wild type hair cells

(Fig. 3m–p). These results indicate that in contrast to the PCP mutant *vangl2*, individual hair cell PCP is normal in Wnt pathway mutants. These findings suggest that the concentric arrangement of hair cells in Wnt pathway mutants is caused by a PCP-independent mechanism.

**The Wnt and PCP pathways have different temporal requirements.** The differences in the hair cell orientation phenotypes of Wnt and PCP pathway mutants suggest that they do not act in

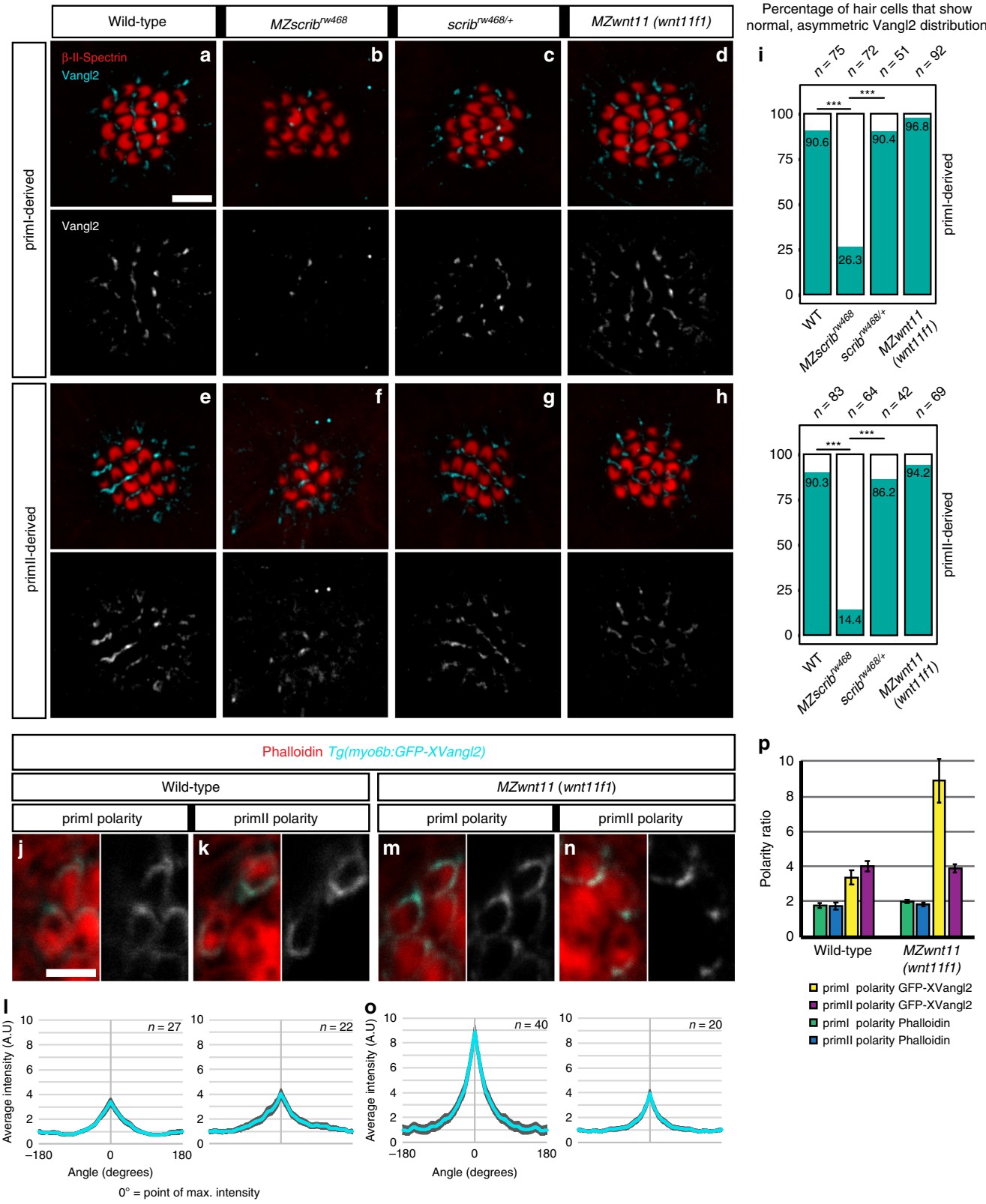

Percentage of hair cells that show normal, asymmetric Vangl2 distribution

0° = point of max. intensity

GFP fluorescence around the circumference (360°) of the cuticular plate

the same pathway. To identify further differences in their phenotypes we investigated PCP-dependent cell behaviors of hair cell progenitors in vivo. The two daughters of a hair cell progenitor often change their position with respect to each other before differentiating[38,39] (Supplementary Movie 1). These cell rearrangements and their duration depend on functional PCP signaling during development and regeneration. In *vangl2* mutants the hair cell rearrangements do not lead to complete reversal of positions, they sometimes occur multiple times and their duration is prolonged[39]. In contrast, time-lapse analyses of the behavior of developing hair cell progenitors in primII-derived neuromasts of *MZwnt11 (wnt11f1)* mutants reveals no significant

**Fig. 3** Vangl2 asymmetry is not affected in hair cells of *MZwnt11* (*wnt11f1*) mutants. **a–d** Double β-II-Spectrin (labeling the cuticular plates) and Vangl2 immunodetection in primI neuromasts of 5 dpf wild type (**a**), *MZscrib* mutants (**b**), *scrib* siblings (**c**) and *MZwnt11* (*wnt11f1*) mutants (**d**). **e–h** Double β-II-Spectrin (labeling the cuticular plates) and Vangl2 immunodetection in primII neuromasts of 5 dpf wild type (**e**), *MZscrib* mutants (**f**), *scrib* siblings (**g**) and *MZwnt11* (*wnt11f1*) mutants (**h**). **i** Quantification of the percentage of hair cells that show asymmetric Vangl2 for each of the conditions in both primordia-derived neuromasts. **j–n** Clonal localization of GFP-XVangl2 in hair cells of neuromasts that show primI and primII polarity in wild type (**j**, **k**) and *MZwnt11* (*wnt11f1*) mutants (**m**, **n**) neuromasts. The majority of the clones were localized in neuromasts in the head, which show the same hair cell orientation as in primI and primII-derived neuromasts in each of the cases. **l**, **o** Quantification of GFP-XVangl2 fluorescence around the cuticular plates (360°) of hair cells in neuromasts that show primI and primII polarity of wild type (**l**) and *MZwnt11f* mutants (**o**). **p** GFP fluorescence and Phalloidin enrichment (polarity ratio, calculated from the cap of max value to the lowest value) in the different conditions analyzed in (**l**, **o**). Error bars in **l**, **o**, and **p** represent S.E.M. ***Fisher's exact test $p < 0.01$. Scale bar in **a** equals 5 μm, **j** equals 2 μm

differences in the number of times they rearrange, the duration of the rearrangements or the angle of divisions with respect to the radius of the neuromast compared to wild type (Supplementary Fig. 4a–f). We conclude that hair cell pairs arise normally in *MZwnt11* (*wnt11f1*) mutants. We also identified differences in the timing of the onset of the hair cell phenotypes between PCP and Wnt pathway mutants. In wild type larvae a pair of hair cells develops from a common progenitor[38] with the kinocilia of both hair cells pointing towards each other in 96% of the cases (Fig. 4a). The first pair of hair cells in *MZwnt11* (*wnt11f1*) mutants shows correct orientation 69% of the time (Fig. 4b), while in *vangl2* mutants they orient properly only in 23% of the cases (Fig. 4c). We conclude that the majority of *MZwnt11* (*wnt11f1*) mutant hair cells possess the proper orientation with respect to their sister hair cell as they form, whereas most *vangl2* mutant hair cells show defective hair cell polarities that are not coordinated with their sister hair cell as soon as the hair cell progenitor divides.

As *MZwnt11* (*wnt11f1*) mutant hair cells eventually all show orientation defects with respect to their sister hair cell, we asked if this phenotype arises progressively. We measured the polarity of the first hair cell pair over time as additional hair cells appear and observed that in wild-type neuromasts the hair cells keep pointing towards each other or their polarity is even refined (Fig. 4d; Supplementary Movie 2). In contrast, in *MZwnt11* (*wnt11f1*) mutants the correct orientation is progressively lost (Fig. 4e; Supplementary Movie 3). In *vangl2* mutants, the aberrant polarity of the first pair also changes over time (Fig. 4f; Supplementary Movie 4). At this point, we do not understand if this effect of cell crowding on the Wnt and PCP phenotypes is caused by the same mechanism or not.

As 5 dpf *vangl2* hair cells do not arrange in concentric circles, we asked whether concentricity might be an intermediate state that is subsequently lost in PCP mutants. We performed time-lapse analyses of developing hair cells in *vangl2* and *MZwnt11* (*wnt11f1*) neuromasts between 4 and 5 dpf. *vangl2* hair cells never arrange in a concentric fashion and are always randomly oriented (Fig. 4g, h; Supplementary Fig. 4g). Thus, concentricity is not a property of developing PCP-deficient hair cells at any stage of their development. Likewise, 4 dpf *MZwnt11* (*wnt11f1*) (Fig. 4i) mutant hair cells are not organized in a concentric fashion, even though they are misaligned (Supplementary Fig. 4g). However, starting at 5 dpf, concentricity is detected in *MZwnt11* (*wnt11f1*) mutants (Fig. 4j; Supplementary Fig. 4g). Therefore, concentricity arises over time in *MZwnt11* (*wnt11f1*) mutants as more hair cells are being added. We conclude that the hair cell defects in PCP and Wnt pathway mutants are caused by different mechanisms, because the majority of *vangl2* mutant hair cells are already misaligned with respect to their sister hair cell after progenitor division, whereas the misalignment of the two sister hair cells in *MZwnt11* (*wnt11f1*) mutant hair cells arises over time.

**Wnt pathway genes coordinate support cell organization**. As the hair cell defect arises over time in *MZwnt11* (*wnt11f1*)

mutants we wondered if the hair cell phenotype could be secondary to defects in neighboring support cells. Support cells surround and sit underneath developing hair cells and serve as hair cell progenitors[38,60,61]. Interestingly, support cells are aligned along the A-P and D-V axes in primI and primII-derived neuromasts, respectively, raising the possibility that the Wnt pathway genes might be acting on them (Fig. 5a, b).

To better visualize any defects in support cell organization we removed hair cells by soaking the larvae in the antibiotic neomycin[62] and calculated their orientation with respect to the horizontal (Supplementary Fig. 5a). In wild-type larvae, support cells show a coordinated, elongated alignment along the horizontal plane (A-P axis) in primI-derived neuromasts (Fig. 5c) and vertical plane (D-V axis) in primII-derived neuromasts (Fig. 5d). These cell alignments demonstrate that hair cell ablation does not disrupt instrinsic support cell alignment. Unexpectedly, we observed that in *vangl2* mutants primI-derived support cells are normally aligned along the horizontal axis, even though their hair cell orientation is randomized (Fig. 5e). *vangl2* mutant support cells in primII-derived neuromast still align, but offset by 90° horizontally (Fig. 5f), a finding we currently cannot explain. Thus, support cells in primII-derived neuromasts of *vangl2* mutants show a cell orientation defect, but not randomization. This suggests the *vangl2* might play some not understood role in only primII-derived support cell orientation. However, the *MZscrib* PCP mutant shows wild type support cell organization in both primI (Fig. 5g) and primII-derived neuromasts (Fig. 5h). These results imply that functional PCP might not be required for coordinated support cell alignment but that PCP genes act mostly in hair cells. On the other hand, in the Wnt pathway mutants *MZwnt11* (*wnt11f1*) and *gpc4*, primI-derived support cells are horizontally aligned (Fig. 5i), whereas support cells in primII-derived neuromasts do not possess an evident coordinated vertical organization (Fig. 5j; Supplementary Fig. 5b). To investigate a possible genetic interaction between PCP and Wnt pathway genes during the establishment of support cell organization, we asked whether the support cell phenotype in Wnt pathway mutants would disrupt the support cell alignment in PCP mutants. Double *vangl2;MZwnt11* (*wnt11f1*) homozygous mutants indeed show a loss of alignment in primII-derived neuromasts and normal, horizontal alignment of support cells in primI neuromasts as characteristic for single *MZwnt11* (*wnt11f1*) mutants (Fig. 5k, l). This result shows that the *wnt11* (*wnt11f1*) phenotype in support cells is epistatic and disrupts normal support cell alignment in PCP mutants.

To test if the support cell phenotype in *wnt11* (*wnt11f1*) mutants is secondary to the hair cell phenotype or vice versa we prevented hair cell development in wild type and *wnt11* (*wnt11f1*) mutants by depleting the hair cell specifying transcription factor *atoh1a*[63] by CRISPR/Cas9-mediated genome editing (Supplementary Fig. 5c, d). While the uninjected wild-type siblings possess functional hair cells that can be labeled with the actin-binding protein β-II-Spectrin (Fig. 5m, n; Supplementary Fig. 5e),

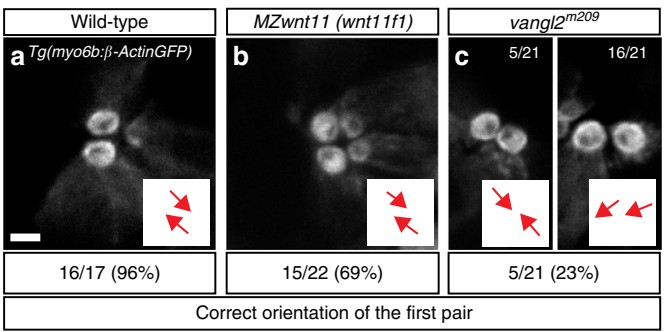

**Fig. 4** Differences in the appearance of the hair cell phenotypes in *vangl2* and *MZwnt11 (wnt11f1)* mutants. **a–c** Orientation of the first pair of hair cells in primII-derived neuromasts of wild type (**a**), *MZwnt11 (wnt11f1)* mutants (**b**), and *vangl2* mutants (**c**) labeled with *Tg(myo6b:β-Actin-GFP)*. In **c**, the left panel shows the number of correctly oriented pairs, and the right panel shows the number of incorrectly oriented pairs (the majority). The percentages for each of the conditions represent the number of pairs that show correct hair cell orientation. **d–f** Hair cell orientation over time in wild type (*n* = 3 animals) (**d**), *MZwnt11 (wnt11f1)* mutants (*n* = 5) (**e**), and *vangl2* mutants (*n* = 2) (**f**) labeled with *Tg(myo6b:β-Actin-GFP)*. Each of the conditions is depicted using a color-coded cartoon, and the hair cell orientation of the first pair is represented by arrows in the box on the lower right. To differentiate between each hair cell of the pair, each of the arrows is in a different color. **g–j** Hair cell orientation of the same *vangl2* mutant fish at 4 dpf (**g**) and 5 dpf (**h**), and the same *MZwnt11 (wnt11f1)* mutant embryo at 4 dpf (**i**) and 5 dpf (**j**) in a *Tg(myo6b:β-Actin-GFP)* transgenic background. The distribution of angles of each given hair cell with respect to the nearest ellipse tangent (concentricity) for primII-derived neuromasts of all the conditions tested is shown. Note that only the *MZwnt11 (wnt11f1)* mutants at 5 dpf (**j**) show significant concentricity (Uniform distribution *p*-values in g = 0.223, h = 0.469, i = 0.308, j = $1.97 \times 10^{-07}$). The von Mises distribution is shown for visual display of the data distribution. Yellow lines represent the polarity axis for each hair cell. "Not Concentric" vs. "Concentric" labels in **g–j** were based on the statistical significance (*n* = 8 neuromasts for each condition). Scale bars in **a**, **d**, and **g** equal 2 μm

wild-type *atoh1a* CRISPants show no hair cells (Fig. 5o, p). In *atoh1a* CRISPants support cells are still normally aligned (Fig. 5o, p). Thus, support cell alignment is independently regulated from hair cells. Furthermore, *MZwnt11 (wnt11f1)* mutants injected with *atoh1a* CRISPR still show loss of support cell alignment in primII-derived neuromasts (Fig. 5q, r). Thus, the disorganization

of support cell orientation observed in primII neuromasts of Wnt pathway genes is not a consequence of the hair cell misorientation. Altogether, these results indicate that, while *vangl2* and *scrib* are not required for support cell alignment, the Wnt pathway genes are essential for the proper coordinated organization of support cells in primII-derived neuromasts.

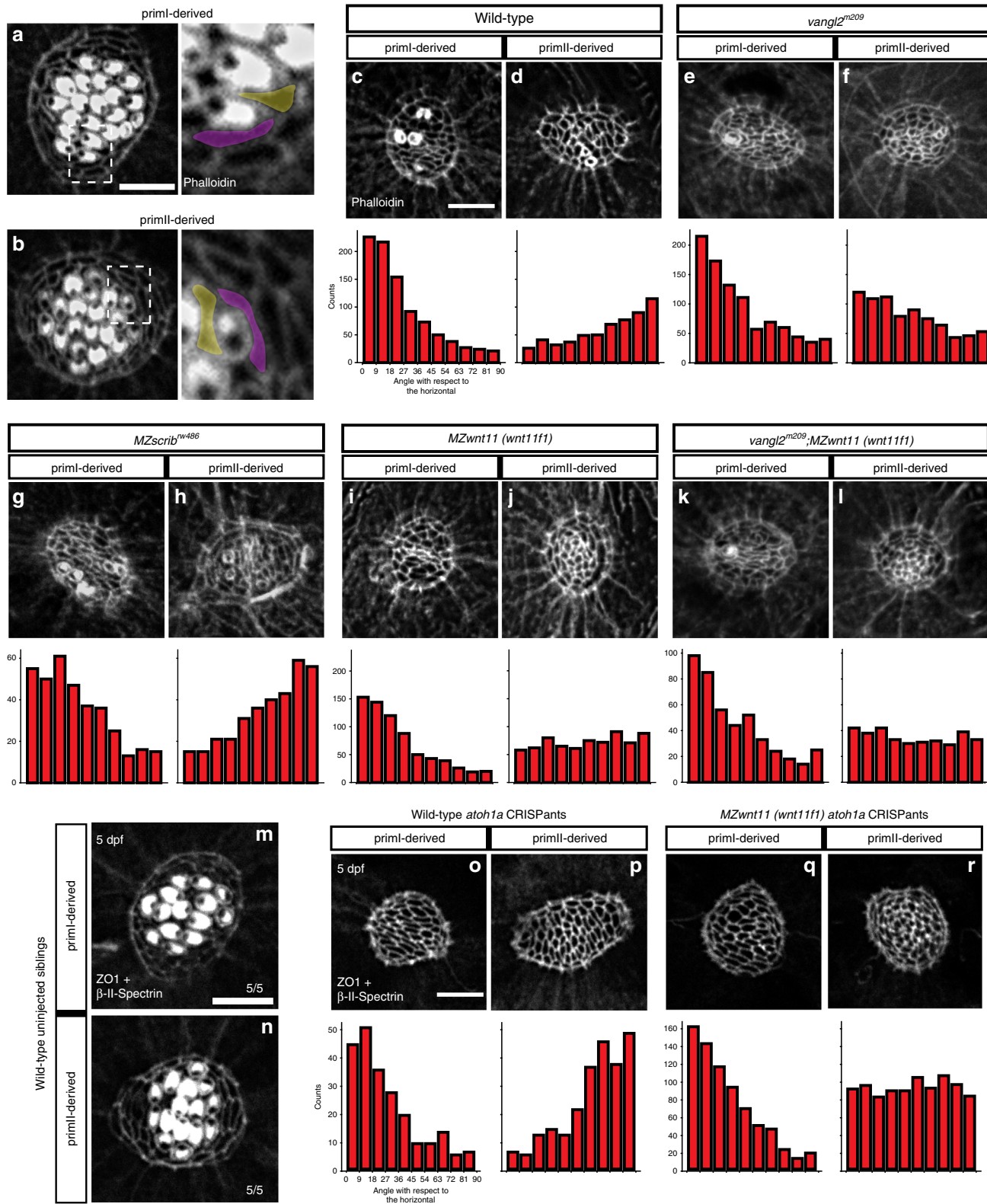

**Wnt pathway genes likely act in the premigratory primordium.** Because *wnt11 (wnt11f1)* is not expressed in 5 dpf primII neuromasts, which show a phenotype in *MZwnt11 (wnt11f1)* mutants (Figs. 1 and 5), we hypothesized that *wnt11 (wnt11f1)* might be required earlier in lateral line development to establish hair cell orientation. We tested if Wnt pathway members are expressed in primII or in adjacent tissues during migration or placode specification stages. At 50 hpf, while primII is migrating and has not

yet deposited its first neuromast, *vangl2* is expressed in primII, and in primI and primII-derived neuromasts, suggesting that it is acting within lateral line cells (Fig. 6a). Likewise, the other PCP gene *scrib* is detected in the lateral line by in situ hybridization, agreeing with a previous RNA-Seq analysis[64] (Fig. 6b). In contrast to *vangl2*, *wnt11 (wnt11f1)* mRNA is not detectable in either the migrating primordia, or primI-derived neuromasts (Fig. 6c). However, it is expressed in the underlying muscle along the

**Fig. 5** Wnt pathway genes are required for proper coordinated support cell organization. **a**, **b** Phalloidin staining of wild-type primI and primII-derived neuromasts. The dotted square outlines the area magnified in (**a**) and (**b**). Two support cells each were false colored to depict their orientation. **b–j** Phalloidin staining of neuromasts 3 h after hair cell ablation. **c**, **d** Wild-type primI (**c**) and primII-derived (**d**) neuromasts. **e**, **f** vangl2 mutant primI (**e**) and primII-derived (**f**) neuromasts. **g**, **h** MZscrib mutant primI (**g**) and primII-derived (**h**) neuromasts. **i**, **j** MZwnt11 (wnt11f1) mutant primI (**i**) and primII-derived (**j**) neuromasts. **k**, **l** vangl2;MZwnt11 (wnt11f1) double mutant primI (**k**) and primII-derived (**l**) neuromasts. The histograms show the distribution of binned angles of cell orientation with respect to the horizontal for each of the conditions tested. **c**, **d** Angle distribution in wild-type primI (**c** one-way chi square test $p = 1.78 \times 10^{-87}$) and primII-derived (**d** $p = 4.54 \times 10^{-19}$) neuromasts. Note that primI distribution is skewed toward being horizontally aligned, while the distribution in primII-derived neuromasts is toward vertical alignment. **e**, **f** Angle distribution in vangl2 mutant primI (**e** $p = 4.91 \times 10^{-13}$) and primII-derived (**f** $p = 2.14 \times 10^{-16}$) neuromasts. **g**, **h** Angle distribution in MZscrib mutant primI (**g** $p = 1.40 \times 10^{-14}$) and primII-derived (**h** $p = 9.60 \times 10^{-13}$) neuromasts. **i**, **j** Support cell orientation in MZwnt11 (wnt11f1) mutant primI (**i** $p = 5.32 \times 10^{-52}$) and primII-derived (**j** $p = 0.006$) neuromasts. **k**, **l** Support cell orientation in vangl2; MZwnt11 (wnt11f1) double mutant primI- (**k** $p = 6.69 \times 10^{-26}$) and primII-derived (**l** $p = 0.22$) neuromasts. **m**, **n** Double β-II-Spectrin (labeling the cuticular plates) and ZO-1 staining of a 5 dpf wild-type neuromasts. **o**, **p** Wild-type atoh1a CRISPants do not possess hair cells (β-II-spectrin is absent; see also Supplementary Fig. 5) and the support cells in all neuromasts show coordinated orientation (one-way chi-square test $p$-value in o = $4.98 \times 10^{-20}$, p = $2.63 \times 10^{-19}$). **q**, **r** MZwnt11 (wnt11f1) injected with atoh1a CRISPR show coordinated support cell orientation in primI-derived neuromasts (**q** $p = 2.88 \times 10^{-60}$) and do not show coordinated support cell orientation in primII neuromasts (**r** $p = 0.24$). Scale bar in **a**, **c**, and **o** equal 5 μm

myoseptum[20,65], suggesting that the muscle could signal to the neuromasts (Fig. 6d; Supplementary Fig. 6a). In contrast to the wnt11 (wnt11f1) ligand, the Wnt co-receptor gpc4 is expressed in the migrating primII and primI-and primII-derived neuromasts, even though its loss only affects primII-derived neuromasts (Fig. 6e, Fig. 1). The expression of the fzd7a and fzd7b receptors is more complex. Both are expressed in the lateral line, but while fzd7a is expressed in primII, and primII-derived neuromasts (Fig. 6f), fzd7b is not expressed in primII but is expressed in primI-derived neuromasts (Fig. 6g). As only fzd7a is expressed in the migrating primII but only fzd7a/7b double mutants affect primII-derived neuromasts, we hypothesize that Wnt signaling is already acting during premigratory stages.

Analysis of the expression pattern of the Wnt pathway genes at a time point in which primII is being specified (32 hpf) shows that wnt11 (wnt11f1) is expressed in a dispersed group of cells posterior to the ear immediately adjacent to the forming, gpc4-expressing D0 placode (Fig. 6h-j arrowhead). In addition, wnt11 (wnt11f1) is expressed in the lens, brain, ear and dorsal spinal cord. The D0 placode gives rise to primII, primD (that gives rise to the dorsal lateral line)[25,34] and the occipital prim (occipital lateral line) (Fig. 6j arrowhead).

Time-lapse analysis of the formation of primII in wild type embryos shows how primII, primD and the occipital prim arise from the D0 placode between 24 and 48 hpf (Fig. 6k–n; Supplementary Movie 5). Neuromasts in all these three lateral lines possess the same D-V hair cell orientation in wild-type embryo (Supplementary Fig. 6c), and also show concentric hair cell organization in the Wnt pathway mutants (Supplementary Fig. 6c). Due to the fact that the neuromasts derived from primD and the occipital prim do not migrate over any wnt11 (wnt11f1)-expressing muscle cells, unlike primII in the trunk, we conclude that the hair cell defect in primII neuromasts is not due to loss of Wnt signaling from muscle cells along the horizontal myoseptum. The only time these three lateral lines are all exposed to wnt11 (wnt11f1) is when they arise from D0, suggesting that wnt11 (wnt11f1) is required in the placode before the three primordia migrate into different directions (Fig. 6o, p).

Single-cell RNA-sequencing (scRNA-Seq) analysis of neuromasts also suggests that the Wnt pathway genes act in support cells, rather than hair cells[66]. gpc4 and fzd7a/b are robustly expressed in support cells but downregulated as cells differentiate into hair cells (Fig. 6q; Supplementary Fig. 6b). vangl2 and scrib are expressed in both support and hair cells (Fig. 6r; Supplementary Fig. 6b), however, the loss of vangl2 only affects the hair cells. The role of vangl2 in support cells is therefore still unresolved.

Overall, these results suggest that while PCP genes are expressed and required to establish proper hair cell planar polarization within all neuromasts, Wnt pathway genes act earlier in support cells during D0 placode formation, prior to the differentiation of hair cells. Support cells then secondarily influence hair cell orientations in 5 dpf primII neuromasts (Fig. 6p, q). Thus, the loss of Wnt signaling early in development has effects on organ formation several days later.

## Discussion

PCP and Wnt pathway genes are involved in the establishment of hair cell orientation in vertebrates[23,24,54–56,67–71]. Because disruptions of both pathways cause hair cell arrangement defects, Wnt pathway mutants are often classified as PCP mutants. Wnt ligands play different roles in different PCP contexts. For example, Wnt ligands instruct PCP[15–17,44] or act as gradients that control hair cell orientation in the inner ear[23,24]. Our results demonstrate that in the context of neuromasts, the Wnt pathway genes wnt11 (wnt11f1), gpc4 and fzd7a/7b do not affect PCP signaling as Vangl2 localization is normal in Wnt pathway mutants (Fig. 3). In addition, in MZwnt11 (wnt11f1);vangl2 double homozygous neuromasts, hair cell orientations are randomized as in vangl2 mutants, suggesting that PCP signaling is normal in MZwnt11 (wnt11f1) mutants (Fig. 2). Recent reports show that Emx2 and Notch signaling control stereocilia bundle orientation and that Vangl2 is normally localized in emx2 mutants suggesting that this pathway acts in parallel to the PCP pathway[36,72]. As emx2 affects hair cell polarity without affecting PCP signaling, we wondered if the Wnt pathway genes acted in concert with Emx2 or Notch signaling. However, the number of Emx2-expressing hair cells is normal in MZwnt11 (wnt11f1), even though they are not localized to one half of the neuromasts but are randomly distributed (Supplementary Fig. 7a–e). Therefore, Wnt pathway genes are not acting in the same pathway with Notch and Emx2 signaling to regulate hair cell orientation. Instead, our data suggest that Wnt pathway genes act in emx2-negative support cells, whose misalignment causes hair cell orientation defects secondarily in the absence of Wnt signaling.

An unresolved question is why in Wnt pathway mutants the concentric hair cell phenotype arises sequentially as more cells are added. A concentric orientation might represent the most energy-efficient arrangement for hair cells that still possess functional PCP but that are surrounded by misaligned support cells. Our results stress the importance of studying all cell types in an organ when analyzing hair cell phenotypes in vertebrates. Therefore, gpc4 and MZfz7a/7b mutants previously characterized as PCP signaling mutants in other contexts should be re-evaluated as such.

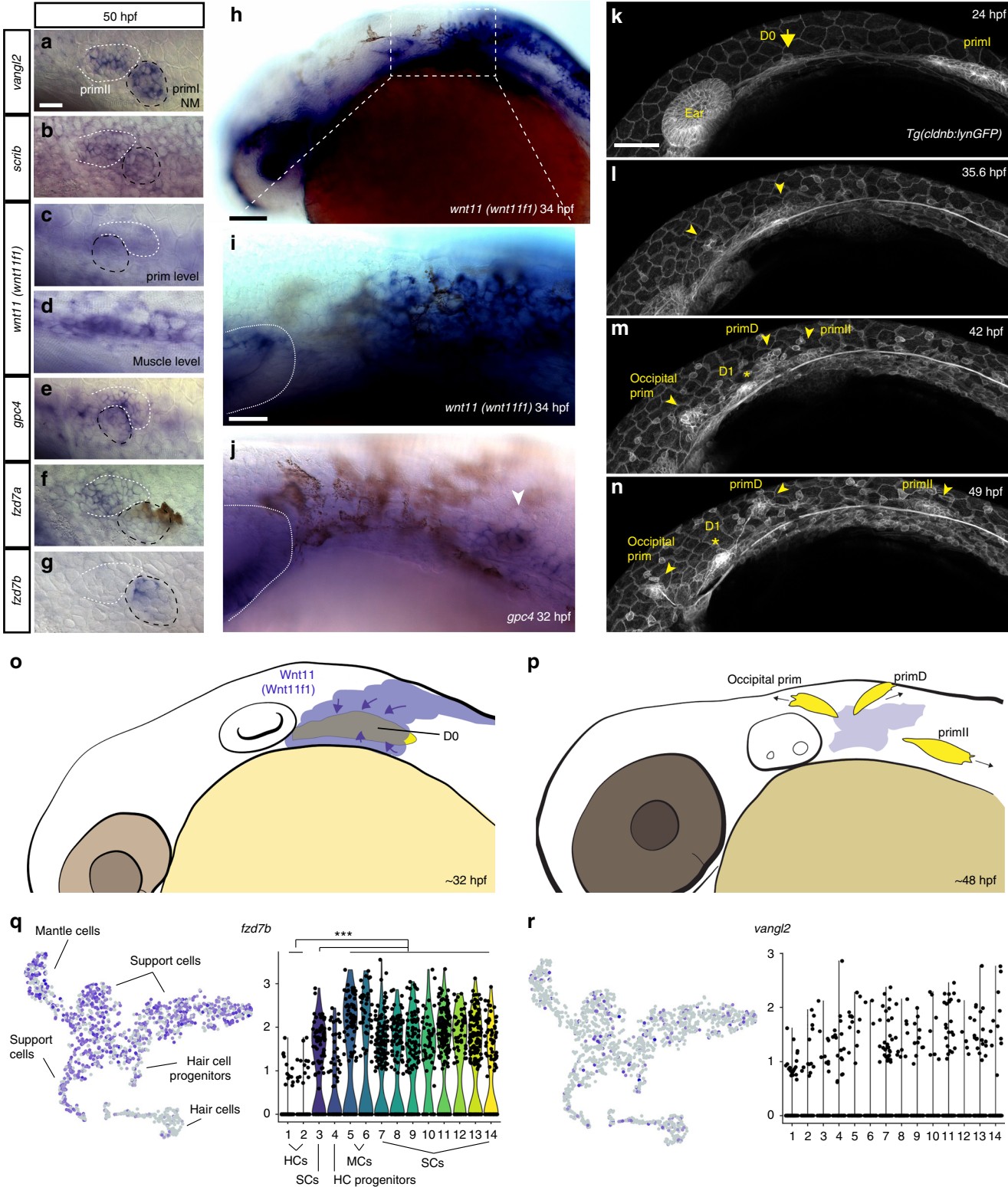

How are the different orientations in primI and primII-derived neuromasts achieved? PCP pathway mutations disrupt hair cell orientations in all neuromasts. Thus, the mechanisms that control differential A-P and D-V orientations in primI and primII-derived neuromasts are either acting upstream or in parallel to PCP signaling. Lopez-Schier et al.[37] proposed that the direction of the migrating primordia or the recently deposited neuromasts determines the axis of polarity. Neuromasts deposited by primII migrate ventrally after deposition[32,73], which has been correlated

with instructing the axis of hair cell polarity with a 90° angle with respect to primI-derived neuromasts. However, the adjacent primI-derived interneuromast cells also undergo a ventral migration and form neuromasts with primI-polarity[25]. Likewise, the D1 neuromast forms from cells left behind by primI and possesses primI polarity (Supplementary Fig. 6c), even though it undergoes a dorsal migration after being formed[25]. Therefore, the direction of migration is not an indicator of hair cell polarity. Our data suggest that support cell polarity in different primordia is set

**Fig. 6** Temporal dynamic expression of the PCP and Wnt pathway genes during formation and migration of primII. **a–g** In situ localization on 50 hpf wild-type fish of mRNA for *vangl2* (**a**), *scrib* (**b**), and *wnt11* (*wnt11f1*) (**c**), at the level of the primordium; (**d**), at the level of the underlying muscle), *gpc4* (**e**), *fzd7a* (**f**), *fzd7b* (**g**). The first neuromast derived from primI is outlined using black, while primII is outlined in white in (**a–g**). **h** In situ localization of *wnt11* (*wnt11f1*) mRNA in a 34 hpf wild-type embryo. **i** Magnification of the expression in the area posterior to the ear in (**h**). The ear is delimited by a dashed line. **j** In situ localization of *gpc4* mRNA in the area posterior to the ear at 32 hpf. Arrowhead indicates the putative localization of the D0 placode. **k–n** Still frames of the time lapse analysis of the formation of primII, primD, occipital prim, and D1 labeled in a *Tg(cldnb:lynGFP)* transgenic wild-type fish (Supplementary Video 5). Arrow in **k** indicates the original group of cells that gives rise to all three primordia. Arrowheads in **l–n** indicate the different primordia formed. Asterisk in **m**, **n** indicates the position of the D1 neuromast. **o**, **p** Schematic cartoon of the proposed mechanism by which Wnt11 (Wnt11f1) signals to cells in the D0 placode (**o**) to establish support cell organization (**p**). **q** t-SNE plots and violin plots showing expression of *fzd7b* (**q**) and *vangl2* (**r**) in a 5 dpf neuromast during homeostasis. The Wilcox *p*-value for q = $8.45 \times 10^{-32}$ in support cells and mantle cells versus hair cells, while in **r** the expression pattern was too sparse to generate meaningful statistics. HCs hair cells, SCs support cells, MCs mantle cells. *** denotes statistical significance. Scale bar in **a** equals 10 μm, **h** equals 50 μm, **i** equals 25 μm, **k** equals 50 μm

up as they are formed, and which instructs hair cell polarity in later forming neuromasts.

Wnt pathway genes are only required for support cell organization in primII- and not primI-derived neuromasts (Fig. 5) raising the question of how primI-derived neuromasts are polarized. Support cells in primI neuromasts could be organized by a mechanism that involves tissue tension as described in the skin of mouse[74], ciliated epithelia of Xenopus[75,76] and in Drosophila[77,78]. Alternatively, primI-derived neuromasts may rely on a different subset of Wnt ligands, HSPGs or Fzd receptors to organize their support cells[79,80]. To date, the analysis of hair cell orientation in zebrafish mutations in two noncanonical Wnt ligands, *wnt5b* (*pipetail*) and *wnt11f2* (formerly known as *wnt11/silberblick*[40]), revealed no hair cell orientation defects in any primordia (Supplementary Figs. 1 and 7f).

Our data suggest that tissue organization in the primordium is set up early during development and influences hair cell orientation in later stages. This mechanism is conceptually analogous to the ratchet effect proposed by Gurdon et al.[81] and implies that cells have a memory of the initial conditions or signals to which they have been exposed that modulates a later response. A similar process is at work during zebrafish gastrulation, where prolonged cadherin-dependent adhesion initiates Nodal signaling and feeds back to positively increase adhesion and determines mesoderm vs. endoderm fate[82]. In addition, reports from cells remembering the spatial geometry or PCP that precedes cell division using the extracellular matrix or tricellular junctions support this hypothesis[83–85]. The memory of early polarity establishment might explain why primII-derived Wnt pathway mutant neuromasts possess a phenotype, even though *wnt11* (*wnt11f1*) is not expressed in surrounding tissues at that stage (Figs. 1 and 5).

Rather than acting as an instructive cue, Wnt pathway genes might be acting as permissive factors by controlling cell–cell adhesion. *wnt11* (*wnt11f1*) destabilizes apical cell–cell junctions in the epithelium during pharyngeal pouch formation in zebrafish[86], and expression of dominant-negative Wnt11 in Xenopus shows that Wnt11 controls cell adhesion through cadherins[87]. Interestingly, *wnt11f2* interacts with *gpc4* and *fzd7a/7b* and influences E-cadherin-dependent cell–cell adhesion, rather than PCP during zebrafish CE and eye formation[19,50,88]. However, even though *wnt11* (*wnt11f1*) binds to the same receptors as *wnt11f2*, *wnt11f2* does not affect hair cell development, suggesting that the two *wnt11* orthologs likely control different adhesion molecules in the different tissues.

Another intriguing possibility is that the Wnt pathway genes might act through the Fat-Dachsous (Ft-Ds) pathway. In flies, disruptions in Ft-Ds signaling cause swirling wing hair patterns and disruptions of the global alignment of PCP proteins without affecting their asymmetric distribution, reminiscent of the phenotypes we observed in Wnt pathway mutants[3,6,89–92]. Furthermore, a recent report shows that the Ft-Ds pathway directs the uniform axial orientation of cells in the Drosophila abdomen[93],

which may be similar to the coordinated organization observed in support cells in our study. It is likely that *wnt11* (*wnt11f1*), *gpc4* and *fzd7a/7b* establish support cell organization by controlling their adhesive properties. However, the presence of a large number of adhesion molecules, and potential for functional compensation, makes functional analyses challenging.

In summary, we identified and characterized the Wnt pathway genes *wnt11* (*wnt11f1*), *gpc4* and *fzd7a/7b*, and the PCP gene *scrib* as distinct inputs controlling hair cell orientation in the lateral line of zebrafish. The core PCP components *vangl2* and *scrib* control individual hair cell polarity, while the Wnt pathway genes *wnt11* (*wnt11f1*)/*gpc4*/*fzd7a/7b* affect the alignment of support cells, which are the progenitors of the later forming hair cells (Fig. 7). This interpretation is supported by our findings that PCP mutants show hair cell orientation defects in the presence of normally aligned surrounding support cells. Conversely, in Wnt pathway mutants support cells are misaligned, even in the absence of hair cells. Importantly, scRNA-Seq analysis of neuromasts shows that *gpc4*, *fzd7a* and *fzd7b* are robustly expressed in support but not hair cells, whereas *vangl2* is also strongly expressed in hair cells. In addition, because the affected support cells in Wnt pathway mutants are only exposed to *wnt11* (*wnt11f1*) ligand while the primordia are still in the head region, we propose that Wnt pathway genes act early during development before the hair cells appear.

Our study provides an alternative mechanistic model in which hair cell orientation defects are not only caused by disruptions of the PCP pathway but independently also by other pathways, such as Wnt11 (Wnt11f1)-activated Wnt signaling. The precise molecular mechanism by which Wnt signaling affects hair cell orientation is unknown but it possibly affects adhesion between support cells required to establish or maintain hair cell alignment.

## Methods

**Fish husbandry.** This study was conducted in accordance with the recommendations in the Guide for the Care and Use of Laboratory Animals of the NIH and a protocol approved by the Institutional Animal Care and Use Committees of the Stowers Institute (TP Protocol: # 2017-0176).

All experiments were performed per guidelines established by the Stowers Institute IACUC review board. The following mutant fish strains previously described were used: *tri*[m209 94], *gpc4*[fr6 45], *wnt11* (*wnt11f1*)[fh224](*wnt11r*[fh224] in[95]), *wnt11f2*[tz216](*wnt11/slb* in[96]) *fzd7a*[e3] and *fzd7b*[hu3495 97], *scrib*[rw468 98], *wnt5b*[ti265 58].

The following transgenic fish lines were used: *Tg(cldnb:lynGFP)*[zf106 99], *Tg (myo6b:β -Actin-GFP)*[100], *Tg(cxcr4b:H2A-EGFP)*[101], *Et(krt4:EGFP)*[sqET4] and *Et (krt4:EGFP)*[sqET20 102], *Tg(cldnB:H2A-mCherry)*[psi4 103].

**In situ hybridization.** Samples were fixed in 4% PFA overnight at 4 °C. Samples were then dehydrated in PBS/Tween 0.3% (PBSTw) and a Methanol series (0, 25, 50, 75 and 100%) for 10 min/each and kept overnight at −20 °C in 100% Methanol. Then, embryos were rehydrated in PBSTw and a Methanol series and incubated with Proteinase K (10 μg/mL in PBSTw) for 2 min at RT. The fish were then postfixed in 4% PFA in PBS for 30 min. After postfixation, the larvae were incubated in Hybridization mix for 2 h at 65 °C, and then were hybridized with the Probes overnight at 65 °C. The following day the unbound probes was removed using a series of washes (50% Formamide/50% 2× SSC; 50% Formamide/5× SSC/

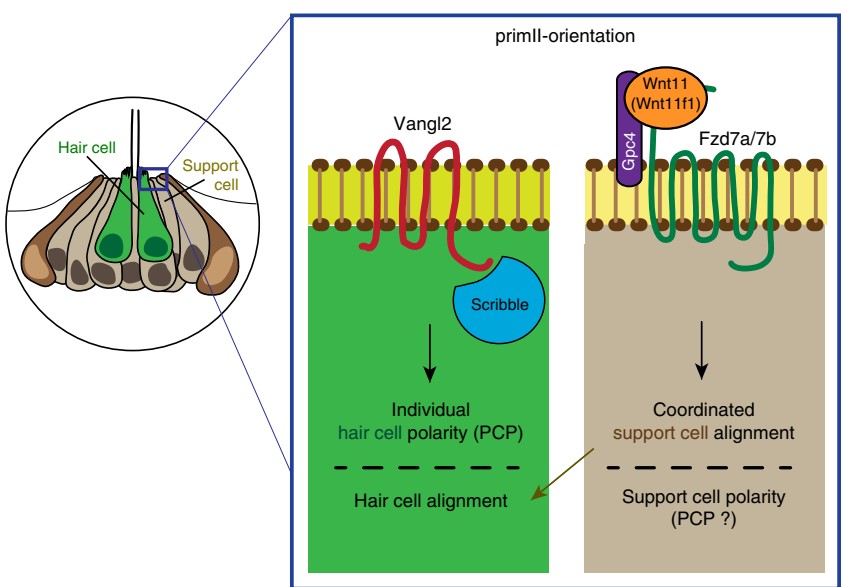

**Fig. 7** The PCP and Wnt pathways act in parallel to establish coordinated hair cell polarity and orientation in the neuromast. Model of how the PCP signaling genes Vangl2 and Scrib are required to establish individual polarity in the hair cells (left part of the box), while the Wnt pathway members Wnt11 (Wnt11f1), Gpc4 and Fzd7a/7b interact to establish coordinated support cell alignment (right part of the box) in primII-derived neuromasts. The coordinated alignment of support cells, which we assume have normal polarity because coordinated alignment is not disrupted in PCP signaling mutants, then dictates the individual hair cell orientation

0.25% CHAPS (Hyb Wash); 50% Hyb wash/50% 2× SSC; 25% Hyb wash/75% 2× SSC; 2× SSC/0.25% CHAPS; 0.2 SSC/0.25% CHAPS; PBSTw/0.25% CHAPS; 100 mM Maleic Acid/50 mM NaCl pH adjusted to 7.5 (MAB); MAB/0.1% Tween) using an Intavis InSituPro robot and detected with an alkaline phosphatase-bound anti-DIG Fab fragment (Roche) (1:4000 in MABT + 10% NGS + 1 mg/mL BSA). Colorimetric development was performed using nitrotetrazolium blue (NBT) and bromochloroindoyl phosphate (BCIP)[104,105]. The following probes were used: *wnt11 (wnt11f1)* (called *wnt11r* in[65]), *gpc4*[106], *fzd7a* and *fzd7b*[107]. The *vangl2* probe was generated using the following pair of Fw/Rv primers (TCATCTCAGAAGCG ATCCAGTA/CCACATGCATCAGACCTAAAAA), the *scrib* probe was generated using the following pair of Fw/Rv primers (AGGATCTTGCCAAGCAAGTG/ CCCTGGCGTAGTTTACGTTT), and the *vangl1* probe was generated using the following pair of Fw/Rv primers (AGCCCTGCTTCTCTCTATGTGT/ CAAACTTGTGTGATTTGGGATG) from wild-type AB DNA.

**Immunohistochemistry and Phalloidin staining**. For β-II-Spectrin, ZO-1 and Vangl2 antibody stainings, embryos were fixed in 2% trichloroacetic acid (TCA) in water at room temperature for at least 4 h. Embryos were then washed briefly in 0.5% Triton-X in PBS (PBSTx) and then blocked for 2 h in PBSTx + 2% normal goat serum (NGS). Primary antibodies Rabbit anti-Vangl2 (1:100, Anaspec, AS-55659s, now discontinued), Mouse anti ZO-1 (1:200, Invitrogen #339100), Mouse anti β-II-Spectrin (1:200, BD Transduction, 612562) were diluted in Blocking solution and incubated overnight at 4 °C. Primary antibody was washed using PBSTx and then detected using either Alexa-568 Anti-Rabbit and Alexa-488 Anti-Mouse Secondary antibodies for the double β-II-Spectrin/Vangl-2; or Alexa-594 Anti-Mouse (Thermo Fisher) for the double β-II-Spectrin/ZO-1 diluted 1:500 in Blocking Solution, and incubated overnight at 4 °C. Secondary antibody was washed thoroughly the next day using PBSTx before imaging.

For Emx2 immunolabeling 5 dpf embryos were fixed in 4% PFA overnight at 4 °C. Then, embryos were permeabilized in acetone for 5 min at −20 °C and incubated with Blocking Solution (2% NGS, 1% BSA in PBS) for 2 h. Fish were then incubated with primary antibody (rabbit anti-Emx2, K0609; Trans Genic, Japan; 1:250) overnight at 4 °C. The following day, fish were washed in 1× PBS and incubated with the secondary antibody (1:250 in Blocking Solution) for 2 h at RT. The fish were then washed thoroughly in PBS and imaged[72].

For Phalloidin staining, embryos were fixed in 4% PFA in 1× PBS at room temperature for at least 2 h. PFA-fixed embryos were briefly washed in 0.5% Triton-X in 1× PBS and then permeabilized in 2% PBSTx for 2 h at room temperature. Subsequently, the embryos were stained with Alexa Fluor 488 Phalloidin (Thermo Fisher; 1:40 in PBSTx 0.5%) for 2 h. After staining, the embryos were washed thoroughly using PBSTx 0.5% and imaged.

**Neomycin treatment**. To kill hair cells, 5 dpf embryos were treated with 300 μM Neomycin in 0.5 E2 media for 30 min[62]. Neomycin was then washed away using fresh E2 media, and waited 3 h before fixation in 4% PFA in 1× PBS.

**Time-lapse imaging and in vivo imaging**. Embryos were anesthetized with Tricaine (MS-222) and mounted in 0.8% low-melting point agarose in glass bottom dishes (MatTek, USA). Embryos were imaged with a Zeiss 780 confocal microscopes using a 40×/1.1 W Corr M27 objective in a climate-controlled chamber set to 28 °C[27].

**Hair cell orientation analysis**. To calculate the hair cell orientation from the Phalloidin-stained neuromasts, the Line tool from FIJI[108] was used. A straight line was drawn from the pole opposite to the kinocilium towards the pole were the kinocilium was found. To create the Rose diagrams to represent hair cell orientation, angles measured and plotted using the rose.diag() function in R ggplot. Statistical analysis was performed for each condition against wild type using Fisher's exact test. To determine A-P or D-V binomial distribution, the data are binned to four groups, left, right, bottom, and up. The binomial test is used to test whether 'right + left' counts are different from 50% of total counts.

**Measure of hair cell alignment and concentricity**. We must choose a structure against which to compare the cell's angles. Specifically, we are looking at relative differences in the angles and an assumed underlying arrangement, rather than comparing them to some absolute feature, such as the horizontal. The first comparison we consider is the angular difference between nearest neighbors. Note although it is possible to provide a consistent polarization of the cell and, thus, measure the angle between two nearest major axes on a scale of (−180, 180)° we are not currently interested in the polarization, thus, we always take the smallest angle between the two major axes. Hence, the angles will be on a scale of (−90, 90)°. Further, the sign convention we will be using is positive if the angle measured from the current cell is anticlockwise, and negative otherwise. The second comparison we consider is cell alignment when compared to a fitted ellipse. Initially, we take the cell centers of the Lines calculated in the "Hair cell orientation analysis" section, and use a least squares fitting algorithm based on principal component analysis to create an ellipse of best fit. Each cell is then projected to the closest point on the fitted ellipse's boundary and their major axis angle is compared to that of the ellipse's tangent at the nearest point. The smallest angle is extracted using the above sign rule. The "Nearest neighbor measurement" provides us with a measurement of how aligned each cell is with its neighbors, while the "Nearest tangent measurement" will provide us with a measurement of how aligned each cell is with the fitted ellipse. Specifically, this comparison will suggest whether there is a circular structure underlying the cell alignment. Having derived the data from the experimental results we plot the information as a histogram. Specifically, the range (−90, 90)° is divided up into 12 bins and we tally how often each angle falls within one of these bins. Dividing by the total number of results then normalizes the histograms to represent a probability. Each angle distribution was first tested against the null hypothesis that there was no preferred direction. Thus, we compare the probability histogram against the uniform distribution and see if there was a significant difference, namely p < 0.01. If there was a significant difference we then compared the solutions to the Von Mises distribution, which is the generalized

Normal distribution that is periodic on the angular domain. The fitting of the Von Mises distribution tells us if there is a preferred direction and how variable the alignment is. The code for these analyses is publicly available at https://github.com/ThomasEWoolley/Cell_alignment.

**Vangl2 asymmetry quantification**. Vangl2 asymmetry quantifications from immunolabeled embryos were performed from raw images. Upon visual inspection, Vangl2 being "asymmetric" vs. "not being asymmetric" was assigned and quantified using the Cell Counter function of FIJI. Fisher's exact test was performed to determine statistical differences between samples. From the hair cells that showed Vangl2 asymmetry, a second analysis was performed and classified into three categories: we quantified whether the enrichment was (1) in the pole where the kinocilium is, (2) on the opposite pole, or (3) enriched but out of the axis determined by the kinocilium. Fisher's exact test was performed to determine statistical differences between samples. The double immunolabeling images were processed for contrast and sharpness only for visual display in the Figure afterwards using Photoshop CS6.

**Generation of the *Tg(myo6b:GFP-XVangl2)* construct**. GFP-XVangl2 was amplified from pDest2-GFP-Vangl2 plasmid (A kind gift from J Wallingford[109]) and Topo TA cloned. The Topo vector was then digested using KpnI and SacI, and ligated into the KpnI-SacI sites of the Tol2 middle entry vector[110] using Quick ligase (NEB). The middle entry GFP-XVangl2 vector was recombined via a Gateway reaction with the 5′ myo6b vector[100]. The final myo6b:GFP-XVangl2 plasmid was co-injected into AB or MZwnt11 (wnt11f1) mutant embryos together with Tol2 Transposase mRNA at a final concentration of 30 ng/µl.

**GFP-XVangl2 profile analysis**. Images were acquired in a Zeiss 780 microscope using a 40×/1.1 W Corr M27 objective. Raw images of Phalloidin-stained animals were used for the analysis of GFP intensity. GFP-XVangl2 polarity was determined in a manner analogous to[111]. These utilized custom plugins as described above. Average cortical line profiles were generated with a 4 pixel thickness utilizing the "polyline kymograph jru v1" plugin. Then maximum angular position of intensity was determined using the "trajectory statistics jru v2" plugin. Profiles were then aligned so that the maximum intensity occurred at angle 0 with "set multi plot offsets jru v1". Next, angles were wrapped to values between −179 and +180 degrees with "wrap angle profile jru v1". Finally, profiles were averaged with "average trajectories jru v1" (available at http://research.stowers.org/imagejplugins/) with error bars indicating the standard error in the mean. For some presentations, averaged profiles were normalized to a maximum intensity of 1 for comparison purposes. Peak widths were determined by fitting profiles to a Gaussian peak function via nonlinear least squares fitting (plugin is "fit multi gaussian jru v1") with error bars determined by fitting 100 Monte Carlo simulated random data sets[112]. Polarity ratios were determined by averaging three values surrounding the 0°, 90°, and 180° data points and calculating their ratios with errors propagated according to ref. [112].

**Hair cell progenitor angle of division analysis**. Imaging was performed in a Zeiss 780 using a 40×/1.1W Corr M27 objective, under conditions described above (Time-lapse imaging section). To obtain a quantitative measurement of how hair cell progenitors divide within the neuromast, we acquired time-lapses of migrating primII in wild type and MZwnt11 (wnt11f1) mutant embryos in a tg(cxcr4b:H2A-EGFP); tg(myo6b:β -Actin-GFP) double background, scanning every 5 min. We calculated the division angle of the dividing progenitor with respect to the radius from the center of division, to the center of the neuromast.

We used Imaris ×64 9.2.1 to manually mark the location of all cell divisions and the center point of the depositing neuromast for all timepoints. To find the center of the neuromast we create a Surface object using the H2A-EGFP channel with very coarse surface detail (surface grain size ~ 20 µm). For each cell division, we used Spots to manually mark the center of each daughter hair cell. The Surface and Spots data for every timepoint is exported and aggregated in an Excel spreadsheet. With the exported data in Excel, we calculated two vectors for every dividing cell pair. We calculated the vector between each of the daughter cells, as well as the vector from the center of the neuromast to the center of cell division. The smallest angle between the two vectors is calculated to give the division angle with respect to the radius, which is the range from 0° to 90°. Division angles for Wild type and mutant were then plotted and t -test analysis analysis performed to calculate statistical differences between wild type and MZwnt11 (wnt11f1) mutants.

**Support cell orientation analysis**. To obtain the support cell orientation, Phalloidin-stained or ZO-1-stained neuromasts were processed in FIJI[108]. A custom "gray morphology jru v1" plugin (available at http://research.stowers.org/imagejplugins/) was used to sharpen the signal from the cell membranes and afterwards segmented using the MorphoJ plugin[113]. Once a mask of the cells within the neuromast was obtained, we found the orientation of each cell using FIJI's "Analyze Particles" function. This function fits each cell to an ellipse in order to provide spatial data, including the center point, major and minor axis length, and the major and minor axis angle in cartesian coordinates. We exported the data from FIJI to Python where we processed and plotted the data. We used Matplotlib to create a histogram of the distribution of cell orientation angles in the range of

0°–90°. To visualize and verify the data, we also plotted the major axis vector on top of the cell mask image for each sample. The code for this analysis is publicly available at https://github.com/richard-alexander/neuromast_cell_orientation.

***vangl1* CRISPR**. We designed two guide RNAs using CRISPRscan (www.crisprscan.org)[114] targeting exon 3 of vangl1 (ENSDARG00000004305). The guide-RNA plus PAM sequences are 5′-GGTTCCAGTGACCGCCGTGGTGG-3′ and 5′-GGCTTTGGGAAGGACACGGAGGG-3′ in vitro transcription using a MEGA-Shortscript T7 kit(Invitrogen) was carried out the manufacturer's recommended conditions. RNA was purified using an RNA Clean-up and Concentrator kit (Zymo Research). One-cell stage embryos were injected with an injection mixture containing 1 µg of Cas9-NLS Protein (PNABio) with 170 ng of each guide-RNA. To assess cutting efficiency, the region around the cutting sites was amplified from genomic-DNA by PCR using GGAGACCTGGGTACTTCCAT and TTAGCAGATCTCATCAAGAACA as primers using GoTaq Master Mix (Promega) and run on an agarose gel.

***atoh1a* CRISPR**. To target atoh1a (ENSDARG00000055294), CHOPCHOP tool[115] was used to design the target sites. The protocol described in[116] was followed to generate two guide RNAs targeting atoh1a's single exon. The generic DNA Oligo B AAAAGCACCGACTCGGTGCCACTTTTTCAAGTTGATAACGGACTAGCCT TATTTTAACTTGCTATTTCTAGCTCTAAAAC was annealed with DNA oligos possessing a T7 promoter plus atoh1a's coding regions 5′-TAATACGACTCAC TATAGGCTGGCTCCCGTGCAGGCGTTTTAGAGCTAGAAATAGC-3′ and 5′-TAATACGACTCACTATAGGGAGAGGCGAAGAATGCAGTTTTAGAGCTAG AAATAGC-3′ and amplified by PCR. From the resulting DNA template, the in vitro reaction was carried out with the T7 MEGAshortscript kit (Invitrogen) under manufacturer conditions, and the RNA was purified and concentrated using the RNA Clean & Concentrator-25 kit (Zymo, #R1017). A mixture of 30 ng of each gRNA was mixed with 1 µg of recombinant Cas9-NLS protein (PNA BIO, CP04-100) and injected into one-cell stage embryos. To assess the success of the injections, 5 dpf CRISPant embryos were stained with DASPEI (2-(4-(dimethylamino) styryl)-N-ethylpyridinium iodide, [Invitrogen, USA]) diluted in embryo media and screened for the lack of hair cells compared with the uninjected siblings. To assess cutting efficiency, the region around the cutting sites was amplified from genomic-DNA by PCR using ATGGATGGAATGAGCACGGAT and AGTTTCAGTTCCG ACAGCTCG as primers and run on an agarose gel.

**Single-cell RNA-Seq**. The scRNA-Seq has been described in detail in[66]. The BAM files and count matrices produced by cell ranger can be accessed in the Gene Expression Omnibus (GEO) database (accession number GSE123241).

Together with this paper we published a publicly available, fully searchable database that allows the user to determine in which neuromast cell type candidate genes are expressed: https://piotrowskilab.shinyapps.io/neuromast_homeostasis_scrnaseq_2018/.

**Reporting summary**. Further information on research design is available in the Nature Research Reporting Summary linked to this article.

## Data availability
The authors declare that all data supporting the findings of this study are available within the article and its supplementary information files or from the corresponding author upon reasonable request. The source data underlying Figs. 1,2,3,4 and Supplementary Figs. 2, 4 and 7 are provided as a Source Data file. Uncropped blots for Supplementary Figs. 2 and 5 are also provided in the Source data file. This has also been deposited in the Stowers Institute Original Data Repository and available online at http://www.stowers.org/research/publications/libpb-1424.

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

## Acknowledgements

We would like to thank Drs. R. Krumlauf, M. Lush, and S. Baek for comments on the paper and the Piotrowski lab members for lively discussions. We also thank Dr. R. Duncan for the initial *wnt11* (*wnt11f1*) in situ analysis, the Aquatics Facility of the Stowers Institute for Medical Research for their outstanding husbandry work. We grateful to T. Nicolson for sending us the *myo6b:β-Actin-GFP* fish, C. Walsh for the *scribble* mutants, D. Slusarski for the *wnt5b* mutants, and J. Wallingford for the *Xenopus* GFP-Vangl2 construct; and L. Solnica-Krezel, A. Chandrasekhar, M. Harris, R. Burdine, B. Ciruna, C.P. Heisenberg, and B. Link for reagents used in pilot studies. Many thanks also to D. Diaz for help with the bioinformatic analysis. This work was funded by an NIH award RO1 NS082567 to C.M., an NIH (NIDCD) award 1R01DC015488-01A1 to T.P and by institutional support from the Stowers Institute for Medical Research to T.P. J.N. A. is a Predoctoral Researcher at the Graduate School of the Stowers Institute for Medical Research.

## Author contributions

J.N.A. designed the experiments, collected, analyzed, and interpreted the data, and wrote the paper; R.L.A., T.W., J.R.U., and H.L. analyzed the data; C.M. provided the *wnt11* (*wnt11f1*)^*fh224*^ mutants; M.G.V. discovered the hair cell orientation phenotype in *wnt11* (*wnt11f1*) mutants. T.P. designed and interpreted the experiments, and wrote the paper.

## Additional information

**Competing interests:** The authors declare no competing interests.

