## [Peer Review File · Nature Communications]

Reviewers' Comments:

Reviewer #1:

Remarks to the Author:

Acedo et al. investigate how Wnt signaling and planar cell polarity (PCP) proteins contribute to the establishment of hair cell orientation in zebrafish lateral line primordia. The authors show that mutations in the selected Wnt and PCP genes result in different outputs as manifested by the hair cell orientation. The mutations in *vangl2* and *scrib*, presumed core PCP genes, cause random orientation of hair cells, whereas *wnt11f1*, *fzd7a/b*, and *gpc4*, considered in the manuscript as the 'Wnt pathway' genes, are manifested as a concentric pattern of hair cells. Based on these loss-of-function experiments, the authors conclude that Wnt and PCP proteins function as separate parallel pathways to establish hair cell orientation. Additionally, *wnt11f1* mutations appear to affect support cell orientation long before the formation of hair cells.

The work described in the present manuscript is solid and delivers valuable information to the field. Specifically, a novel 'concentric phenotype' is described for *wnt11f1* mutants. Moreover, the authors demonstrate that *wnt11f1* mutation likely causes the early misalignment of support cells rather than hair cells, providing a new mechanistic insight. Nevertheless, since any particular mutation may be masked by redundant gene activity, the insights obtained from lack of expected phenotypes do not fully exclude a role of Wnt signaling in neuromast polarity, compromising the authors' conclusions. There is an additional concern that the selection of the Wnt pathway components and PCP players has not been sufficiently justified. In summary, the manuscript in its current form does not provide a significant conceptual advance to the field.

Other concerns:

- Why is *scrib* selected as a 'core' PCP gene and used despite not being detected in the lateral line by *in situ* hybridization (Figs. 6a, b)? Rather, loss-of-function experiments for *prickle*, *celsr*, or *disheveled* would appear more valuable and appropriate. Conversely, the authors should explain why *fzd7a/b* is classified as a non-PCP gene, contrary to the well-known role of *fz* in *Drosophila* PCP.
- The data for the double mutant (MZ*wnt11f1fh224*; *vangl2m209*) could be added to Fig. 4, shedding more light on Wnt and PCP pathway analysis.
- The study would be considerably improved by biochemical read-outs in neuromasts, such as JNK activation or Vangl2 mobility shift assay (Gao et al., *Dev Cell*, 2011), to confirm the authors' claims and *in vivo* relevance.
- The scRNAseq study appears unpublished and should be better described in Results and Methods.

Minor concerns:

- It is critical to know whether the effect of *gpc4* and *fzd7* mutants on support cell orientation has been confirmed.
- Although *wnt11f1* is expressed in *primI* cells (Fig. 1d, Suppl. Fig. 1), only the *primII* cells are affected in the mutant (Fig. 2). Together with the partial effect of *wnt11f1* reported in Fig. 4b, this observation suggests that *wnt11f1* functions redundantly with another *wnt* gene(s), but this possibility remains to be examined.
- The authors are encouraged to improve the clarity of Fig. 3 and the accompanying text to make it easier to follow.
- The data shown in Suppl. Fig. 8 and Fig. 4g", h", i", j" seem redundant.
- The p-values in Fig. 2 legend do not match the plotted data.
- Fig. 5e'-j' are mentioned in the text (p. 14) but missing in the figures.
- Figs. 6p and 6q are not cited in the text (p. 16, below).
- A few typos need to be corrected: *significative*, *poloarity* (p. 6), *phenoytpes* (p. 12), *stereocolia* (p. 18).

Reviewer #2:

Remarks to the Author:

This paper presents a very thorough treatment of a complex problem and it was a pleasure to read. Sensory hair cells such as those in the inner ear are a key paradigm for understanding the mechanisms of PCP function, and this paper is notable for a) excellent use of the zebrafish neuromasts for this purpose, and moreover, b) raising the bar for how such studies should be carried out. However, while the paper clearly sets the stage for a more penetrating analysis, firm conclusions are hard to come by here. The paper presents a philosophical conundrum for the editor in this age of "mechanism," but it's my view that this is exactly the kind of foundation-building work that deserves attention.

Substantial work has demonstrated the role of PCP genes in hair cell polarization, and has indicated a complex interplay between hair cell and support cells. But progress in this area has been slow. What makes this paper so remarkable, and makes it a very valuable contribution, is its dynamic and very quantitative approach to the complex interplay of hair cells and support cells. By carefully considering the two populations separately and by developing elegant quantification schemes for each, this integrated viewpoint will certainly become state of the art for future studies. In addition, the paper throws some light on the interplay of Wnt and PCP signaling. While it is perhaps not surprising that the situation is more complex than expected, the concentric phenotype that arises in the Wnt mutants is fascinating, and the more so because the patterns of Vangl2 protein localization are unaltered, suggesting that this phenotype, despite appearances, is not related to the whorls observed in PCP mutant fly wings or mouse skin.

Thus, while solid conclusions are scarce, the paper provides a wealth of detailed data that will be both interesting and useful to a wide array of researchers. Importantly, the writing reflects this, and the work does not overstate its case (a minor exception discussed below). I support publication, though I urge the authors to consider the following:

1. I am not convinced that the single cell RNAseq adds value to this paper. First and foremost, without some statistical comparison, I am not fully convinced by the central claim that hair cells downregulate certain genes. This is a shame, because everything else in the paper (including the first half of Fig 7 is important and interesting, and convincing. I recommend saving the scRNAseq for another story.
2. The authors make no mention of Vangl1, using only Vangl2. For cases in which phenotype are obtained, this is not a problem. But for cases in which they observed no phenotypes, it will be essential to know Vangl1 is not involved before claims can be made that the process is PCP-independent. Given recent work from the Deans lab on Vangl1 and vangl2 in the inner ear, this is important.

Reviewer #3:

Remarks to the Author:

The manuscript by Acedo and colleagues describes a novel phenotype, concentric hair cell orientations, in DV oriented lateral line neuromasts, in response to deletion of *wnt11f1*. The figures are outstanding and make it very easy to understand the points that the authors seek to make. The results are intriguing, and the authors very convincingly show that this is phenotype does not occur through the PCP pathway. Unfortunately, the study fails to demonstrate a plausible alternative mechanism to explain the results. The authors demonstrate changes in the orientation of supporting cells, which they suggest could play a role in the polarity defects that arise over time in primII-derived neuromasts, but it is just as likely that the defects in supporting cells arise as a

result, rather than a cause, of the hair cell misorientations. While the demonstration that *wnt11f1* is expressed at the time of early placode formation would seem to pinpoint when the defect arises, the study provides no insights regarding what the nature of the defect might be. This is a significant deficiency in the study as the results then amount to the description of an interesting phenotype with no understanding of how it is achieved, except that one can be confident the PCP pathway is not involved.

Minor comments/questions

Page 7, last line states that there is no additive effect of double deletion of *wnt11f1* and *gpc4*. What would an additive defect look like? *Wnt11* and *gpc4* single mutants already show randomization of orientations.

Page 14, there is a comment that PCP pathway genes are not required for support cells to acquire coordinated organization since orientation is normal in *primI*-derived neuromasts. But isn't there a defect in polarization in *primII*-derived neuromasts in *vangl2* mutants (figure 5f)? And wouldn't this suggest an orientation defect?

Reviewer #1 (Remarks to the Author):

The work described in the present manuscript is solid and delivers valuable information to the field. Specifically, a novel 'concentric phenotype is described for wnt11f1 mutants. Moreover, the authors demonstrate that wnt11f1 mutation likely causes the early misalignment of support cells rather than hair cells, providing a new mechanistic insight. Nevertheless, since any particular mutation may be masked by redundant gene activity, the insights obtained from lack of expected phenotypes do not fully exclude a role of Wnt signaling in neuromast polarity, compromising the authors conclusions.

Response: We would like to thank the reviewer for stating that our work is solid and delivers valuable information. However, we believe it is unlikely that the concentric hair cell phenotype is possibly due to redundancy that masks a random hair cell orientation phenotype. We observed that three genes positioned at different levels in the Wnt pathway possess an identical concentric hair cell phenotype, making it unlikely that different PCP genes act redundantly with each of the three Wnt pathway components. However, we do agree that it is possible that other Wnt ligands, that do not signal via Fzd7 and Gpc4 interact with the PCP pathway. We added this possibility to our Discussion and clarified in the title that Wnt signaling in our manuscript refers to the Wnt11, Gpc4 Fzd7 axis specifically.

There is an additional concern that the selection of the Wnt pathway components and PCP players has not been sufficiently justified.

Response: We analyzed Gpc4 and Fzd7a/b mutants because these two proteins were shown to interact during convergent extension movements (Djiane et al., 2000 Dev.; Ohkawara et al., 2003 Dev.; Rochard et al., 2016 Dev.; Roszko et al., 2015 Dev.; Topczewski et al., 2001 Dev. Cell; Witzel et al., 2006 JCB, page 6 of the manuscript). In addition, it was demonstrated that Wnt11f2 interacts with Gpc4 and Fzd7a/b (Capek et al., 2019; Petridou et al., 2018; Witzel et al., 2006). As an additional core PCP gene besides *vangl2*, we investigated *scribble (scrib)*. Scribble1 interacts with the PCP pathway in the mouse cochlea and its loss causes hair cell defects (Montcouquiol et al. 2003, Montcouquiol et al. 2006). We have added this sentence on page 7.

In summary, the manuscript in its current form does not provide a significant conceptual advance to the field.

Response: We respectfully disagree with the reviewer. Even though we were not able to determine the mechanistic cause of the concentric phenotype, our dynamic and quantitative studies of Wnt pathway mutants show that hair cells can be misoriented because of an underlying support cell phenotype. In addition, in these Wnt pathway mutants the misoriented hair cells still

possess normal PCP. This is in contrast to the randomly oriented hair cells in mutants of core PCP genes. Classically, all hair cell orientation phenotypes have been classified as PCP defects. We were able to detect these differences in hair cell phenotypes because of the small size of the lateral line organs. Such differences in hair cell phenotypes are quite problematic to detect in larger and difficult to access mouse sensory epithelia in which hair cell polarity defects tend to cause very similar looking phenotypes. In addition, our single cell RNASeq analysis shows that Wnt pathway genes are enriched in support cells, whereas core PCP genes are expressed in support and hair cells, again supporting our conclusion that the Wnt pathway genes primarily affect the support cells.

Other concerns:

Why is scrib selected as a 'core' PCP gene and used despite not being detected in the lateral line by in situ hybridization (Figs. 6a, b)? Rather, loss-of-function experiments for prickle, celsr, or disheveled would appear more valuable and appropriate.

Response: The reviewer is correct in questioning why *scrib* was not detected in our original in situ experiments. We have generated a new probe and repeated the in situ hybridization experiments, which now show expression in neuromasts (please see new Fig. 6b). This result is corroborated by the finding that *scrib* is expressed in neuromast cells in scRNASeq experiments (see new Suppl. Figure 6b).

[Redacted]

Conversely, the authors should explain why fzd7a/b is classified as a non-PCP gene, contrary to the well-known role of fz in Drosophila PCP.

Response: We agree with the reviewer that we should discuss in more detail that Fzd7 can play different context-dependent functions (see added text on page 7). We classify *fzd7a/7b* as non-PCP genes in lateral line neuromasts because the PCP pathway is not affected in the mutants and the loss of function phenotype does not resemble a PCP phenotype. However, *fzd7a/7b* might act in the PCP pathway in other organs. For example, during vertebrate dendrite development Fzd7 acts through two different non-canonical Wnt pathways by activating JNK, as well as CamKII (Ferrari et al., J. Cell Sci. 2018). Fzd7 also acts in the canonical and non-

canonical Wnt pathway in *Xenopus* (Medina et al., *Mech. Dev.* 2000) demonstrating that Fzd7 can play different context-dependent functions.

- *The data for the double mutant (MZwnt11f1fh224; vangl2m209) could be added to Fig. 4, shedding more light on Wnt and PCP pathway analysis.*

Response: Unfortunately, we do not have the double mutants in the *myo6b:β-Actin-GFP* background to perform these experiments in an expedient fashion. However, as the double mutants show randomized hair cell polarities at 5dpf (Fig. 2e,j) and the single *vangl2* mutants never show concentric phenotypes during early stages, we believe it is unlikely that double mutant embryos possess a concentric phenotype earlier in development.

- *The study would be considerably improved by biochemical read-outs in neuromasts, such as JNK activation or Vangl2 mobility shift assay (Gao et al., *Dev Cell*, 2011), to confirm the authors' claims and in vivo relevance.*

Response: We agree with the reviewer that analysis of JNK activation or Vangl2 phosphorylation would be informative but performing these experiments with lateral line cells is very challenging. Biochemical analyses would have to be done with FAC sorted lateral line cells. Obtaining a large enough number of lateral line cells for biochemical analyses is almost impossible. Additionally, we currently cannot separate primI from primII-derived neuromast cells and pJNK levels in the lateral line are very low (Drerup and Nechiporuk 2013 *PLoS Genetics*). To circumvent the low expression levels of pJNK in the lateral line, whole-embryo lysates could be used, but this would likely mask subtle differences due to the averaging of expression from other tissues. We believe that using lateral line cells is crucial because signaling can be context-dependent. For example, the Wnt pathway genes act in the PCP pathway in other organs, e.g *gpc4* mutants have a strong PCP-dependent convergent extension defect as well (Marlow et al. 1998 Dev Biol.; Jessen et al. 2002 *Nature Cell Biology* ; Li et al. 2013 Development.; Roszko et al. 2015 *Development*). Wnt11 can even act in the canonical Wnt pathway as described for axis formation in *Xenopus* (Tao et al. 2005, *Cell*). Alternatively, human cells could be used as in the cited Gao paper who used human CHO cells for their mobility shift assay. However, as *wnt11* (*wnt11f1*), *fzd7* and *gpc4* play a role in the PCP pathway in other cell types, we believe using human cells would not be informative as to whether these Wnt pathway genes interact with the PCP pathway in lateral line cells. However, our interpretation that that PCP is unaffected in Wnt pathway mutants is supported by a more recent publication of the Gao group in 2017 who showed that proper Vangl2 localization depends on its phosphorylation. This suggests that in *wnt11* (*wnt11f1*) zebrafish mutants Vangl2 is normally phosphorylated, as it is asymmetrically localized.

- *The scRNAseq study appears unpublished and should be better described in Results and Methods.*

Response: We updated the reference, as the paper is now published (Lush, Diaz et al. eLIFE 2019) and the data is available. We added the corresponding p-values to Figure 6b and Suppl. Fig. 6b). In the eLife paper we used scRNASeq analyses to characterize different cell types in the neuromasts based on their distinct transcriptomes. We described genes that are specific for each cell type (mantle cells, support cells, hair cells and their progenitors) and defined the cascade of gene activation and repression that leads to hair cell differentiation. Together with this paper we published a publicly available, fully searchable database that allows the user to determine in which neuromast cell type candidate genes are expressed.

https://piotrowskilab.shinyapps.io/neuromast_homeostasis_scrnaseq_2018/

We used this web-based application to interrogate in which cell types the Wnt pathway and core PCP genes are expressed and determined that Wnt pathway genes are highly enriched in support cells, whereas core PCP genes are also highly expressed in hair cells. We have added this information to the Method section of the manuscript.

Minor concerns:

• *It is critical to know whether the effect of *gpc4* and *fzd7* mutants on support cell orientation has been confirmed.*

Response: We agree and in response to the reviewer's comment we have now analyzed the support cell alignment in *gpc4* mutants. After hair cell ablation support cells in *gpc4* mutants show a coordinated D-V orientation in primI neuromasts and random orientation in primII-derived neuromasts. This result is identical to the support cell orientation phenotype in the *MZwnt11 (wnt11f1)* mutants, suggesting the Wnt pathway genes act in the same pathway to establish support cell orientation in primII-derived neuromasts (Suppl. Fig. 5b). These new results are described on page 12.

• *Although *wnt11f1* is expressed in primI cells (Fig. 1d, Suppl. Fig. 1), only the primII cells are affected in the mutant (Fig. 2). Together with the partial effect of *wnt11f1* reported in Fig. 4b, this observation suggests that *wnt11f1* functions redundantly with another *wnt* gene(s), but this possibility remains to be examined.*

Response: We believe it is unlikely that *wnt11 (wnt11f1)* is acting redundantly with another Wnt ligand because *wnt11 (wnt11f1)*, *gpc4* and *fz7* mutants, which act at different levels of the pathway, show the same phenotype. Additionally, double *wnt11 (wnt11f1); gpc4* homozygous mutants do not show a defect in primI-derived neuromasts and the concentric phenotype in primII-derived neuromasts is not randomized. We discuss the possibility (page 18) that primI might use a different set of Wnt/s, Gpc/s and Fzd/s to establish support cell orientation.

• The authors are encouraged to improve the clarity of Fig. 3 and the accompanying text to make it easier to follow.

Response: We thank the reviewer for pointing this out. We have rewritten the text on page 9, added information and rearranged Fig. 3j-n' to clarify the description of our results.

• The data shown in Suppl. Fig. 8 and Fig. 4g", h", l", j" seem redundant.

Response: We apologize for the confusion. The data shown in Fig. 4g", h", l", j" refers to the analysis of the angle with respect to the nearest ellipse tangent (concentricity), while the analysis in the new Supplementary Figure 4g refers to the alignment of a hair cells with respect to its neighbor, a measurement of Organization vs Randomness. We have now added titles to the different graphs describing their purpose.

• The p-values in Fig. 2 legend do not match the plotted data.

Response: We thank the Reviewer for spotting this mistake. During the preparation of the Figures and Figure legends, we inadvertently failed to correct the figure legend after deleting a panel. We have corrected the p-values in the Figure Legend (page 22). They now match the data shown in Figure 2.

• Fig. 5e'-j' are mentioned in the text (p. 14) but missing in the figures.

Response: We thank the reviewer for spotting this mistake. We added the letters e'-j back into the Figure.

• Figs. 6p and 6q are not cited in the text (p. 16, below).

Response: We thank the reviewer for pointing this out. References to Figures 6p and 6q have now been added to the text.

• A few typos need to be corrected: significative, pololarity (p. 6), phenoytpes (p. 12), stereocolia (p. 18).

Response: We would like to thank the reviewer for catching the typos.

Reviewer #2 (Remarks to the Author):

Thus, while solid conclusions are scarce, the paper provides a wealth of detailed data that will be both interesting and useful to a wide array of researchers. Importantly, the writing reflects this, and the work does not overstate its case (a minor exception discussed below). I support publication, though I urge the authors to consider the following:

1. I am not convinced that the single cell RNAseq adds value to this paper. First and foremost, without some statistical comparison, I am not fully convinced by the central claim that hair cells downregulate certain genes. This is a shame, because everything else in the paper (including the first half of Fig 7 is important and interesting, and convincing. I recommend saving the scRNAseq for another story.

Response: The scRNAseq paper is now published, including a web-based, publicly available, and fully searchable database (Lush, Diaz et al., 2019, eLIFE). The scRNASeq analysis supports our conclusion that the Wnt pathway genes mainly act in the support cells, as they are hardly expressed in hair cells, whereas the core PCP genes also robustly expressed in hair cells. We would therefore like to report these results in the present manuscript. We have now added the p-values, where applicable, to the panels in Figures 6 and Suppl. Fig. 6.

2. The authors make no mention of Vangl1, using only Vangl2. For cases in which phenotype are obtained, this is not a problem. But for cases in which they observed no phenotypes, it will be essential to know Vangl1 is not involved before claims can be made that the process is PCP-independent. Given recent work from the Deans lab on Vangl1 and vangl2 in the inner ear, this is important.

We thank and agree with the Reviewer. In situ detection of *vangl1* showed that it is expressed in the migrating primI and mantle cells of the recently deposited neuromasts, but not in primII (Suppl. Fig. 2l). Single cell data suggests the expression levels are very low at 5dpf (Suppl. Fig. 2m). Since *vangl1* is expressed in primI and neuromast mantle cells, it is indeed possible that *vangl1* plays a role in hair cell orientation in the lateral line of zebrafish. We functionally tested whether loss of function of *vangl1* by CRISPR injections into wild type embryos affects hair cell orientation (Suppl. Fig. 2n,o). Although we observe body extension defects resembling those caused by defective convergent suggesting *vangl1* plays a role in early development, these embryos do not display any hair cell phenotype (Suppl. Fig. 2p). We assessed the cutting effectivity by genotyping exon 3 in the embryos that displayed the short body phenotype and indeed found that they possessed a complex collection of deletions and insertions and that the *vangl1* gRNAs are effective. We also tested whether loss of *vangl1* function would modify the phenotype of Wnt and PCP pathway mutants. *MZwnt11 (wnt11f1)* mutants into which we injected *vangl1* CRISPR show the same concentric hair cell phenotype as single *MZwnt11 (wnt11f1)* mutants. Thus, the lack of phenotype in *MZwnt11 (wnt11f1)* primI neuromasts or the

concentric arrangement of hair cells in primII-derived neuromasts are not due to compensation by *vangl1*. Finally, hair cells of *vangl2* mutants that were injected with *vangl1* CRISPR do not affect the randomized hair cell orientation phenotype. We therefore suggest that *vangl1* and *vangl2* do not act redundantly in hair cell orientation in the lateral line. We added these results to page 8.

Reviewer #3 (Remarks to the Author):

The manuscript by Acedo and colleagues describes a novel phenotype, concentric hair cell orientations, in DV oriented lateral line neuromasts, in response to deletion of wnt11f1. The figures are outstanding and make it very easy to understand the points that the authors seek to make. The results are intriguing, and the authors very convincingly show that this is phenotype does not occur through the PCP pathway. Unfortunately, the study fails to demonstrate a plausible alternative mechanism to explain the results. The authors demonstrate changes in the orientation of supporting cells, which they suggest could play a role in the polarity defects that arise over time in primII-derived neuromasts, but it is just as likely that the defects in supporting cells arise as a result, rather than a cause, of the hair cell misorientations.

Response: We agree with the reviewer that we originally only presented correlative data suggesting that the hair cell phenotype in Wnt pathway mutants is secondary to support cell defects, rather than the other way around. We now generated data showing that support cells in Wnt pathway mutants are disorganized, even in the absence of hair cells, whereas in wildtype embryos the loss of hair cells does not affect support cell alignment. To test whether support cell defects arise as a consequence of hair cell misorientation defects, we mutated the transcription factor *atoh1a*, crucial for hair cell specification (Itoh and Chitnis 2002 Mech Dev.), using CRISPR (Suppl. Fig. 5). Indeed, *atoh1a* CRISPRants lack hair cells. When *atoh1a* is mutated in wild type embryos, coordinated organization of the support cells is not disrupted, suggesting that support cell alignment is regulated independently of the presence of hair cells (Figure 5m-r'). Additionally, *MZwnt11* (*wnt11f1*) mutants injected with *atoh1a* CRISPR still show disorganized support cell alignment in primII-derived neuromasts, demonstrating that the disorganization of support cell orientation observed in primII neuromasts of Wnt pathway genes is not a consequence of the hair cell misorientation. Together with the fact that in PCP mutant neuromasts (*vangl2* and *scrib*, see Fig. 5) the hair cells orientation is randomized but support cells are properly aligned, we conclude that that support cell orientation is independently regulated from hair cell orientation.

Minor comments/questions

Page 7, last line states that there is no additive effect of double deletion of wnt11f1 and gpc4.

What would an additive defect look like? Wnt11 and gpc4 single mutants already show randomization (not randomization) of orientations.

Response: We agree that this statement does not make much sense. What is more important is that double mutants do not (i) have a hair cell phenotype in primI-derived neuromasts and (ii) that they do not look like a PCP mutant phenotype (*vangl2*, which show random hair cell orientation). Therefore, *wnt11* (*wnt11f1*) and *gpc4* likely act in the same pathway, independent of *vangl2*. We changed the text on page 7 accordingly.

Page 14, there is a comment that PCP pathway genes are not required for support cells to acquire coordinated organization since orientation is normal in primI-derived neuromasts. But isn't there a defect in polarization in primII-derived neuromasts in vangl2 mutants (figure 5f)? And wouldn't this suggest an orientation defect?

Response: We agree with the reviewer that we do not possess solid data that shows that the PCP pathway is not required in support cells and that support cells in primII-derived *vangl2* neuromasts show a support cell alignment/orientation defect (even though they are not randomized and *scrib* mutant support cells are normal). In Fig. 7 we indicate with a '?' that we do not know if PCP is also required in support cells. Accordingly, we now also rephrased the text on page 12.

[Redacted]

Reviewers' Comments:

Reviewer #2:

Remarks to the Author:

My concerns have been addressed and i now support publication in the present form.

Reviewer #3:

Remarks to the Author:

The authors have clearly made a serious effort to address the concerns of the reviewers. Their conclusions are well supported by their data and demonstrate that expression of *wnt11f1* at an early stage of development appears to establish an inherent tissue polarity that is retained by both the hair cells and the supporting cells that arise from those primordial cells. However, in their response to reviewers, they did not address my most relevant comment

"While the demonstration that *wnt11f1* is expressed at the time of early placode formation would seem to pinpoint when the defect arises, the study provides no insights regarding what the nature of the defect might be. This is a significant deficiency in the study as the results then amount to the description of an interesting phenotype with no understanding of how it is achieved, except that one can be confident the PCP pathway is not involved."

Possibly because this issue is still unresolved. The lack of a mechanistic demonstration of how *wnt11f1* plays a role in specification of tissue polarity is not addressed and continues to represent a significant deficiency in the study.